# ING-VP: MLLMs cannot Play Easy Vision-based Games Yet

## Abstract

As multimodal large language models (MLLMs) continue to demonstrate increasingly competitive performance across a broad spectrum of tasks, more intricate and comprehensive benchmarks have been developed to assess these cutting-edge models. These benchmarks introduce new challenges to core capabilities such as perception, reasoning, and planning. However, existing multimodal benchmarks fall short in providing a focused evaluation of multi-step planning based on spatial relationships in images. To bridge this gap, we present **ING-VP**, the first INteractive Game-based Vision Planning benchmark, specifically designed to evaluate the spatial imagination and multi-step reasoning abilities of MLLMs. ING-VP features 6 distinct games, encompassing 300 levels, each with 6 unique configurations. A single model engages in over 60,000 rounds of interaction. The benchmark framework allows for multiple comparison settings, including image-only vs. text-only inputs, single-step vs. multi-step reasoning, and with-history vs. without-history conditions, offering valuable insights into the model's capabilities. We evaluated numerous state-of-the-art MLLMs, with the highest-performing model, Claude-3.5 Sonnet, achieving a best accuracy of only 8.00%, far below the human accuracy of 65.66%. This work aims to provide a specialized evaluation framework to drive advancements in MLLMs' capacity for complex spatial reasoning and planning. The code is publicly available at `https://anonymous.4open.science/r/ING-VP-E49A`.

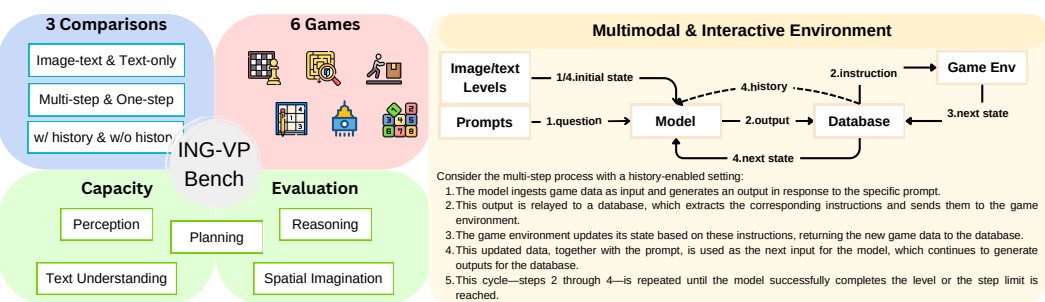

Figure 1: The overview of ING-VP benchmark.ING-VP comprises 6 distinct games, conducts 3 comparative analyses across 6 experimental settings, and evaluates 5 key capabilities of MLLMs. Additionally, it offers a highly efficient interactive environment for both inference and analysis.

## 1 Introduction

Large language models (LLMs) have demonstrated remarkable capabilities in natural language processing, generation, and even textual complex reasoning and planning (Zhao et al., 2023). Building upon this powerful foundation of LLMs, integrating visual inputs has led to the development of even more powerful models (OpenAI, 2024; Anil et al., 2023a), *a.k.a* multimodal large language models (MLLMs).

Despite demonstrating impressive performance in handling most general multimodal tasks, the effectiveness of MLLM in multimodal reasoning and planning still remains unclear. Moreover, recent

studies (Lu et al., 2024; Dai et al., 2024) indicate that vision-language training might degrade the textual capabilities of MLLMs, suggesting that MLLMs built upon LLMs could be impaired when adapted to multimodal reasoning and planning tasks. Consequently, there is an urgent need for a test that incorporates multimodal complex reasoning and planning cases to guide the subsequent enhancements of MLLMs.

To address this issue, existing studies generally utilize visual question answering (VQA) (Antol et al., 2015; Kafle & Kanan, 2017) and game-based evaluations (Wu et al., 2023; Bellemare et al., 2013) to assess the visual reasoning capabilities of MLLMs. In general, VQA necessitates a verified ground-truth answer that relies on human annotations. But acquiring these annotations is both costly and time-consuming. Moreover, the absence of interaction and planning in typical VQA tasks poses difficulties in evaluating the reasoning and planning capabilities of advanced MLLMs. The tasks presented in these benchmarks are overly simplistic (Yue et al., 2023) or only test reasoning within domain-specific knowledge (Yue et al., 2023; Zhang et al., 2024a), which mainly evaluates the LLM knowledge of MLLMs rather than the perception, reasoning, and planning of MLLMs. Therefore, recent studies (Xu et al., 2024; Chia et al., 2024) prompt MLLMs to interact with digital game environments, which are measured by game outcomes and scores, leading to the game-based evaluation. Unlike VQA tasks, these methods can evaluate the multi-step reasoning capabilities and even spatial imagination of MLLMs, which is crucial function of human cognition, allowing us to interact with realistic environments (Wu et al., 2024). Despite the effectiveness, these works are typically restricted to individual games with complex rules, involve time-consuming evaluation episodes, and fail to effectively assess the models' generalization capabilities in multimodal planning. Considering these challenges, our goal is to develop a generalizable and efficient benchmark to evaluate the multi-step planning abilities of MLLMs, providing insights for subsequent improvements of MLLMs with complex multi-step reasoning.

To fill this gap, in this paper, we introduce the **IN**teractive **G**ame-based **V**ision **P**lanning benchmark (ING-VP), meticulously focusing on evaluating the spatial imagination and multi-step reasoning abilities of MLLMs. Figure 1 shows games, evaluation settings, and the interactive process in our ING-VP. To construct our ING-VP, we initially collect six games featuring easily understandable rules. In each game, we collect 50 levels, each comprising both an image and a text representation of the current state, providing vision and textual inputs for MLLMs, as illustrated in Figure 2. To assess the spatial imagination and planning capabilities of MLLMs, we establish six experimental settings, which prompt the models to perform single-step and multi-step reasoning, with or without historical interaction. During the evaluation, we employ MLLMs to interact within the environment until the game is completed. To evaluate model performance comprehensively, beyond merely determining whether a model can finish a game, we also use the model's action efficiency and the remaining steps to complete the game as evaluation metrics.

With our ING-VP, we test 15 open- and closed-source MLLMs and analyze their performance on our test cases. We first support the benchmark designed to evaluate the multi-step reasoning and spatial imagination capabilities of MLLMs — ING-VP bench. Then we analyze these capabilities of current open- and closed-source MLLMs, despite a performance gap, the leading open-source model, InternVL2-Llama3-76B, achieves an accuracy of 2.50%, ranking just behind Claude-3.5 Sonnet, GPT-4o, and Gemini-1.5 Pro. Notably, its performance significantly surpasses that of GPT-4o mini, which stands at 1.05%, and GPT-4v, which records a mere 0.32%. We also conduct a detailed analysis of these models' performance, the evidence shows that:

- The inability to process the relative positions of elements is one of the primary issues with MLLM perception.

- Even the most advanced MLLMs have very limited planning capabilities, far below the performance of ordinary humans on these simple tasks.

- Current models tend to generate instructions that are much longer than necessary to complete the levels. While this can improve accuracy on simple levels, it also indirectly reveals that MLLMs are "uncertain" about the correct solution.

While most tasks in the ING-VP benchmark are straightforward for humans, they pose significant challenges for MLLMs, even the top-performing model, Claude-3.5 Sonnet, achieving an average accuracy of just 3.37%. We reveal that current MLLMs generally lack spatial imagination and multi-step planning abilities, and offer a new perspective on the capability requirements for MLLMs.

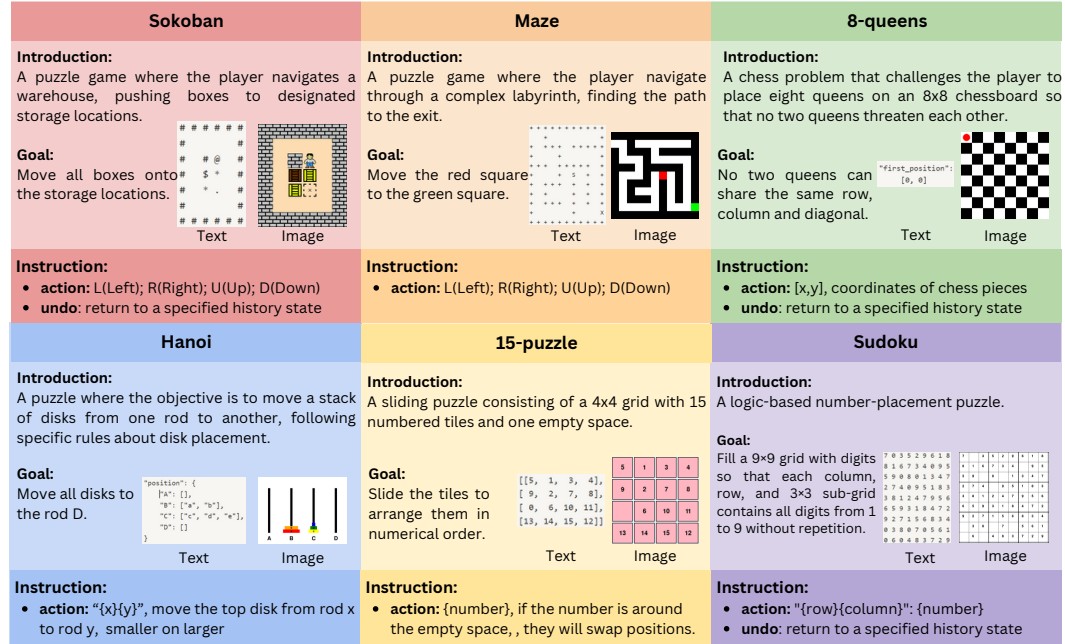

Figure 2: ING-VP examples sampled from each game. Includes pictures and text representations of Sokoban, Maze, Sudoku, 8-queens, Tower of Hanoi, and 15-puzzle.

## 2 RELATED WORK

**Multimodal Large Language Models.** LLMs (Achiam et al., 2023; Anil et al., 2023b) have demonstrated their ability of generating human-like texts to understand and respond to complex instructional queries. The successes of LLMs has elicited the burgeoning proliferation of multi-modal LLMs (Alayrac et al., 2022; Li et al., 2023b; Liu et al., 2024; Sun et al., 2024; Jin et al., 2023b), which is designed to process and integrate multiple types of data. The primary attempt Flamingo (Alayrac et al., 2022) endows visual-language models with in-context few-shot learning capabilities by trained on large-scale interleaved text-image data. BLIP2 (Li et al., 2023b) designs a Q-Former architecture to align the visual-textual knowledge during the pre-training phase. LLaVA (Liu et al., 2024) collect GPT-4 generated multimodal language-image instruction-following data and train a general-purpose visual-language assistant. Beyond multimodal understanding, EMU-2 (Sun et al., 2024) and LaVIT (Jin et al., 2023b) take one step further and act as generative multimodal model to support visual prompting and object-grounded generation.

**MLLM Benchmarks.** The development of MLLMs has highlighted the critical need of benchmarks for thorough evaluations. Although traditional visual-language tasks (*e.g.*, visual question answering (Antol et al., 2015; Kafle & Kanan, 2017) and image captioning (Lin et al., 2014; Plummer et al., 2015)) can be used as evaluation benchmarks, they are too strict and require the exact match with the ground-truth answers. To this end, LVLM-eHub (Xu et al., 2023) and LAMM (Yin et al., 2024) reformulate exiting public datasets as evaluation samples and employ human annotators or GPT to assess the quality. MME (Li et al., 2024), MMBench (Liu et al., 2023b) and SEED-Bench (Li et al., 2024) construct multiple-choice questions to mitigate the subjectivity and instability of GPT evaluation. MMMU (Yue et al., 2024) evaluate the advanced perception and reasoning of MLLMs on specific domains (*e.g.*, science, business).

**Game-based Evaluations.** Digital games are acknowledged as essential in the pursuit of artificial general intelligence since they present complex challenges requiring advanced reasoning and cognitive skills. These challenges make digital games an ideal benchmark for evaluating the capabilities of MLLMs (Wu et al., 2023; Bellemare et al., 2013; Hu et al., 2024; Sweetser, 2024; Xu et al., 2024) including the environment perception (Hong et al., 2023; Akoury et al., 2023), memory construction (Zhu et al., 2023; Zhang et al., 2024b; Ding et al., 2023; Park et al., 2022; Liu et al., 2023a), reasoning (Liu et al., 2023a; Wang et al., 2023a; Qian et al., 2023; Huang et al., 2022) and decision-making

| Benchmark | #Puzzle | Multi-round | Ability | #Metric | Source | Answer |
|---|---|---|---|---|---|---|
| RAVEN (Zhang et al., 2019) | 1 | ✗ | Reasoning | 1 | Synthesized | MC |
| Super-CLEVR (Li et al., 2023c) | 1 | ✗ | Reasoning | 1 | Synthesized | Open |
| ConceptARC (Moskvichev et al., 2023) | 1 | ✗ | Reasoning | 1 | Annotated | Open |
| AlgoPuzzleVQA (Ghosal et al., 2024) | 18 | ✗ | Reasoning | 1 | Synthesized | MC |
| PuzzleVQA (Chia et al., 2024) | 10 | ✗ | Reasoning | 1 | Synthesized | MC |
| COLUMBUS'(Kraaijveld et al., 2024) | 3 | ✗ | Reasoning | 1 | Synthesized, Internet | MC |
| INGVP | 6 | ✓ | Reasoning, Planning | 3 | Synthesized, Internet | Open |

Table 1: Comparison of existing evaluation benchmarks. Compared to other benchmarks, ING-VP employs a multi-round strategy to assess the model's planning capabilities and introduces two additional metrics beyond accuracy to enhance result diversity.

(Chen et al., 2023; Zhou et al., 2023; Jin et al., 2023a; Qian et al., 2023). Several methods focus on semantic-level perception of environmental elements including locations, objects or actions in games. They either use basic text input of user ideas (Li et al., 2023a) or game state variables and dialogues (Akoury et al., 2023; Park et al., 2022; 2023). Role-based inputs, *e.g.*, the inclusion of character, story, role-related information (Hong et al., 2023; Wang et al., 2023b) and skills (Gong et al., 2023) are often included. TorchCraft (Synnaeve et al., 2016) is presented to use real-time strategy games such as StarCraft: Brood War to serve as a benchmark for AI research. The Chess game has long been employed as an AI testing ground (Noever et al., 2020; Stöckl, 2021; Toshniwal et al., 2022). Chess Transformer (Noever et al., 2020) fine-tunes GPT-2 to generate plausible strategies and learns complex gameplay. Recent works (Taesiri et al., 2022; 2024) formulate the bug detection problem as a question-answering task and leverage the zero-shot capabilities of LLMs for video game bug detection. R2-PLAY (Xu et al., 2024) constructs a multimodal game instruction tuning dataset to facilitate the "read-to-play" capability of LLMs. PuzzleVQA (Chia et al., 2024) demonstrates that existing MLLMs exhibit substantial challenges when solving puzzles that demand visual perception, inductive reasoning, and deductive reasoning. Beyond the benchmark setting, we additionally develop an interactive environment to assess the ability of multimodal models to perform spatial reasoning and multi-step inference based on visual details. To further elucidate the distinctions between ING-VP and existing benchmarks, we present the detailed benchmark comparisons in Table 1

## 3 THE ING-VP BENCHMARK

### 3.1 OVERVIEW OF ING-VP

We introduce ING-VP benchmark, a new interactive game-based vision planning benchmark designed to measure the multi-step reasoning and spatial imagination capabilities of MLLMs. The benchmark encompasses 6 distinct settings, 6 games, and 50 levels per game, the core mechanisms are depicted in Figure 1. To mitigate data leakage and ensure problem solvability, the majority of our levels are algorithmically generated and verified. Representative examples of each game are illustrated in Figure 2. Details of the data collection process are provided in the Appendix B.

ING-VP features 6 games that are conceptually simple yet cognitively challenging: Sokoban, Maze, Sudoku, 8-queens, Tower of Hanoi, and 15-puzzle. The simplicity lies in the easily comprehensible rules and the ability to encapsulate complete level information within a single image, facilitating comprehensive reasoning. The challenge stems from the requirement for models to precisely capture core visual elements and their spatial relationships, necessitating multi-step reasoning to successfully complete each level. We meticulously craft 6 reasoning settings, enabling researchers to systematically identify the strengths and limitations of target models through comparative analysis of performance across these settings.

### 3.2 SIX INFERENCE SETTINGS

**One-step: Image and Text-only Settings** In the One-step with Image setting, we provide the model solely with an image depicting the initial game state and prompt it to generate comprehensive instructions for level completion. The One-step Text-only setting follows an identical approach, with the key distinction being the replacement of the image input with its corresponding textual representation.

**Multi-step: Image and Text-only Settings (without History)** In the Multi-step with Image setting, we provide the model with an image of the current game state at each inference round. After the model outputs a single-step instruction, this instruction is fed into the game as input, causing the game state to change and generate a new image. This new image then serves as the model's input for the next step. The Multi-step Text-only setting follows the same process, but uses textual representations as the model's input.

**Multi-step: Image and Text-only Settings (with History)** The key distinction in these settings is the inclusion of the model's historical outputs as part of the prompt in each interaction. Additionally, for Sokoban, Sudoku, and N-queens, we add an undo option, allowing the model to freely revert to any previous state. This enhancement applies to both the Image and Text-only variants of the Multi-step setting.

## 3.3 GAME SELECTION

We chose six games that are widely recognized, have straightforward rules, and operate in a deterministic environment, making them ideal representatives for our study. In a deterministic environment, the outcome of every action taken by an agent is predictable and certain. Such an environment can be formally defined using a Markov Decision Process (MDP). The model employs a strategy $\pi$ to determine the next action $a_t$ based on the current state $s_t$ and all previous actions $a_{0:t-1}$, represented as:

$$a_t = \pi(s_t, a_{0:t-1}) \tag{1}$$

The planning process of MLLMs can be expressed as:

$$S' = \pi(S, A, G, n) \tag{2}$$

Where $S'$ is the future sequence of states, which terminates upon achieving the goal $G$ or exhausting the available moves $n$; $S$ is the current sequence of states; $A$ represents the current sequence of actions.

## 4 EXPERIMENTS

We conduct a comprehensive evaluation of both open-source and closed-source MLLMs, employ a zero-shot setting to faithfully emulate the human puzzle-solving process, given the unique nature of our tasks. A uniform set of prompts was applied across all models. The complete set of 36 prompts is presented in the Appendix C.

## 4.1 BASELINES

**MLLMs.** We consider a comprehensive suite of mainstream large multimodal models. Closed-source models include GPT-4o, GPT-4o Mini, GPT-4v, GPT-4 Turbo, Claude-3.5 Sonnet, Claude-3 Opus, and Gemini-1.5 Pro. Open-source models consist of CogVLM2-19B, DeepSeek-VL, Internvl-Chat-v1.5, Internvl2-8B, Internvl2-26B, Internvl2-40B, InternVL2-Llama3-76B, and MiniCPM-V2.6. We utilize each model's official API for closed-source systems or the publicly available checkpoint for open-source implementations, More information of these models can be found in the Appendix A.

**Evaluation.** We present a systematic interactive environment for evaluating all MLLMs, where models interact with the game environment until either completing the task or exhausting the allotted steps. We constrain the model's output action instructions to JSON format through prompts and extract them using regular expressions. The correctly extracted instructions are then used as input for the game environment. After the game state changes, the new state is fed back to the model for the next round of inference. We employ three metrics: accuracy, completion degree, and action efficiency. (1) Accuracy is our main metric, it measures whether the model can complete the task within the specified number of steps. (2) Completion degree is determined by the final state of the game environment after interaction with the model. The closer the final state is to the cleared state, the higher the score; if it deviates, the score decreases accordingly. (3) Action efficiency represents

whether each instruction output by the model effectuates a change in the game state. The computation method for action efficiency is as follows:

$$\text{Action Efficiency} = \frac{\sum_{i=1}^{n} \frac{\text{\# of efficient actions for level i}}{\text{\# of total actions for level i}}}{n}$$

| Model | Metric | Image-text | | One-step | Text-only | | One-step | Overall |
|---|---|---|---|---|---|---|---|---|
| | | Multi-step | | | Multi-step | | | |
| | | w/o history | w/ history | | w/o history | w/ history | | |
| Closed Source Model | | | | | | | | |
| Claude-3.5 Sonnet | Acc. | 0.30 | 0.30 | 7.00 | 2.30 | 2.30 | 8.00 | **3.37** |
| | Comp. | 3.90 | 4.30 | 21.90 | 4.90 | 5.20 | 16.80 | **9.50** |
| | Eff. | 26.90 | 23.10 | 48.40 | 17.60 | 18.50 | 42.00 | 29.42 |
| GPT-4o | Acc. | 3.30 | 2.00 | 0.30 | 3.30 | 3.30 | 4.30 | 2.75 |
| | Comp. | 6.70 | 5.20 | 12.90 | 5.80 | 5.40 | 13.80 | 8.30 |
| | Eff. | 19.20 | 14.20 | 33.70 | 18.70 | 18.30 | 47.80 | 25.32 |
| Gemini-1.5-Pro | Acc. | 1.00 | 0.30 | 2.70 | 5.70 | 4.30 | 2.30 | 2.72 |
| | Comp. | 5.90 | 3.80 | 9.60 | 8.20 | 6.50 | 8.50 | 7.08 |
| | Eff. | 34.70 | 27.80 | 42.80 | 19.50 | 18.50 | 37.70 | **30.17** |
| GPT-4o mini | Acc. | 0.70 | 0.30 | 0.00 | 2.00 | 2.30 | 1.00 | 1.05 |
| | Comp. | 3.40 | 3.40 | 6.60 | 5.20 | 5.90 | 8.90 | 5.57 |
| | Eff. | 13.20 | 8.20 | 35.20 | 19.50 | 17.30 | 40.10 | 22.25 |
| GPT-4V | Acc. | 0.00 | 0.00 | 1.30 | 0.00 | 0.30 | 0.30 | 0.32 |
| | Comp. | 2.90 | 2.90 | 4.30 | 2.60 | 3.00 | 3.40 | 3.18 |
| | Eff. | 8.80 | 7.20 | 5.50 | 16.80 | 17.40 | 8.50 | 10.70 |
| GPT-4 Turbo | Acc. | null | null | null | 2.30 | 2.30 | 1.00 | 1.87 |
| | Comp. | null | null | null | 4.80 | 4.80 | 9.10 | 6.23 |
| | Eff. | null | null | null | 12.20 | 12.30 | 41.00 | 21.83 |
| Claude-3 Opus | Acc. | null | null | null | 2.30 | 2.30 | 1.00 | 1.87 |
| | Comp. | null | null | null | 4.80 | 4.80 | 10.70 | 5.07 |
| | Eff. | null | null | null | 12.40 | 12.30 | 40.80 | 21.83 |
| Open Source Model | | | | | | | | |
| InternVL2-Llama3-76B | Acc. | 2.67 | 2.33 | 3.00 | 2.33 | 1.67 | 3.00 | 2.50 |
| | Comp. | 9.07 | 6.28 | 8.30 | 8.32 | 8.03 | 5.88 | 7.65 |
| | Eff. | 17.55 | 15.13 | 36.18 | 21.13 | 29.30 | 32.95 | 25.58 |
| Internvl2-26B | Acc. | 2.33 | 1.33 | 1.67 | 1.67 | 2.00 | 2.33 | 1.89 |
| | Comp. | 4.80 | 5.22 | 5.65 | 5.25 | 5.27 | 5.22 | 5.23 |
| | Eff. | 10.58 | 9.22 | 11.93 | 10.22 | 9.27 | 16.72 | 11.32 |
| Internvl2-40B | Acc. | 1.67 | 1.67 | 2.67 | 1.00 | 2.00 | 1.67 | 1.78 |
| | Comp. | 5.68 | 5.43 | 7.87 | 5.03 | 4.08 | 8.08 | 6.03 |
| | Eff. | 18.37 | 12.98 | 22.22 | 15.33 | 15.22 | 34.16 | 18.82 |
| Cogvlm2-19B | Acc. | 1.33 | 0.67 | 2.00 | 1.67 | 1.33 | 2.00 | 1.50 |
| | Comp. | 5.90 | 5.68 | 6.58 | 5.68 | 5.02 | 7.63 | 6.08 |
| | Eff. | 15.75 | 16.45 | 27.12 | 13.75 | 12.85 | 31.37 | 19.55 |
| Internvl2-8B | Acc. | 1.00 | 0.33 | 0.33 | 1.33 | 0.67 | 1.67 | 0.89 |
| | Comp. | 2.60 | 2.58 | 3.33 | 2.63 | 2.50 | 3.83 | 2.91 |
| | Eff. | 5.90 | 5.27 | 4.97 | 3.05 | 4.27 | 6.03 | 4.91 |
| Internvl-Chat-v1.5 | Acc. | 0.67 | 0.33 | 0.00 | 0.33 | 0.33 | 0.67 | 0.39 |
| | Comp. | 6.30 | 6.30 | 4.57 | 5.80 | 6.00 | 4.18 | 5.53 |
| | Eff. | 14.90 | 14.22 | 25.68 | 11.70 | 10.87 | 27.27 | 17.44 |
| deepseek-VL | Acc. | 0.67 | 0.33 | 1.00 | 0.33 | 0.00 | 0.00 | 0.39 |
| | Comp. | 3.47 | 2.72 | 3.65 | 2.68 | 4.18 | 3.92 | 3.44 |
| | Eff. | 11.80 | 11.22 | 16.40 | 8.38 | 9.57 | 15.90 | 12.21 |
| MiniCPM-V2.6 | Acc. | 0.33 | 0 | 0 | 0.67 | 0.33 | 0 | 0.22 |
| | Comp. | 3.78 | 3.33 | 4.17 | 3.62 | 2.68 | 4.22 | 3.63 |
| | Eff. | 11.18 | 10.62 | 17.73 | 10.08 | 6.37 | 21.88 | 12.98 |
| Human | | | | | | | | |
| Human average | Acc. | null | null | 65.66 | null | null | null | 65.66 |

Table 2: Main results for the best-performing MLLMs (LLMs) and humans on different settings.

## 4.2 MAIN RESULTS

In this section, we examine the spatial reasoning and planning abilities of current MLLMs using the ING-VP benchmark. The results are presented in Table 2, please see the Appendix D for the complete results.Our key observations are as follows:

**Huge gap between humans and MLLMs:** Even the most advanced model, Claude-3.5 Sonnet, achieves an accuracy of only 3.37%. By contrast, humans readily achieve an average success rate of 65.66% on these tasks, highlighting a significant gap between model performance and human capabilities on the ING-VP benchmark.

**Performance disparity between open-source and closed-source models persists**: While the performance of closed-source models on ING-VP is far from satisfactory, they still outperform the open-source models. The best-performing open-source model, InternVL2-Llama3-76B, achieves an accuracy of 2.50%, which remains lower than Claude-3.5 Sonnet, GPT-4o and Gemini-1.5 Pro.

**For MLLMs, the greatest challenge in perception is understanding location information.** According to our observations of the inference results, the most advanced models, such as Claude-3.5 Sonnet and GPT-4o, can generally identify the elements present and even count the

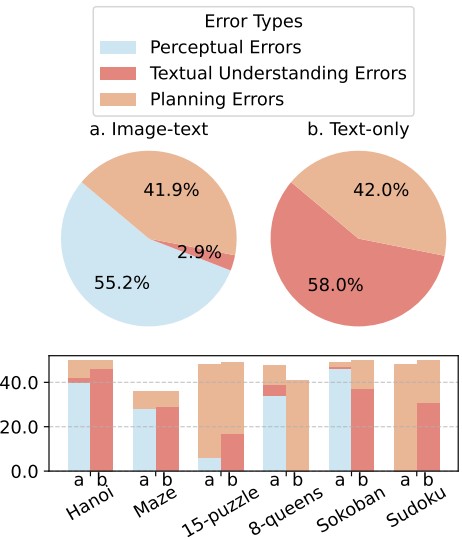

Figure 3: Error distribution over Claude-3.5 Sonnet's 555 errors across different tasks and settings.

quantity of each in the Sokoban game. However, they struggle to accurately determine precise location information, leading to very low inference accuracy and degree of task completion.

**Merely breaking down the steps is unhelpful and may even be counterproductive.** In text-only tasks, Claude-3.5 Sonnet and GPT-4o achieve accuracy rates of 2.30% and 3.30%, respectively, in the multi-step setting, which are lower than their 8.00% and 4.30% accuracy in the one-step setting. For the ING-VP benchmark, thinking step by step does not work and even has a negative effect. We believe that MLLMs rely heavily on pattern matching based on prior training data, generating outputs from similar inputs rather than engaging in actual planning.

## 4.3 FINE-GRAINED ANALYSIS

In this section, We conduct a comprehensive range of analyses to explore the generative capabilities of MLLMs in a broader context, while also dissecting the nuanced output tendencies of current models. We hope our results can provides valuable insights that can inform future model design and training strategies.

**Error Analysis.** We collate and analyze 555 errors (image-text: 279, text-only: 276) made by Claude-3.5 Sonnet in one-step setting, as illustrated in Figure 3. It is important to note that while we categorize each case under distinct error types, in many instances the model exhibited errors in both comprehension and reasoning. Our classification follows contextual cues: when the model provided invalid instructions from the outset, we labele it as an understanding error. Conversely, if the model deviated from the correct solution at an intermediate step, we classify it as a reasoning error. Below, we summarize key observations based on these error types:

- **Perceptual Errors (55.2%/–%):** These errors occur exclusively in the image-text setting. While current models are generally able to recognize overall attributes of an image—such as identifying the game genre and its components, their ability to accurately interpret fine details, including the specific size and precise location of each element, remains limited

(e.g., see Figure 7. This perceptual limitation represents a major contributor to the elevated error rates in this setting.

- **Textual Understanding Errors (2.9%/58.0%):** Textual understanding errors manifest in two main forms: a misinterpretation of specific prompts or an inability to correctly parse data structures or character matrices used to represent game levels in the text-only setting (as shown in Figure 8). These errors indicate that the model struggles to generalize its understanding when presented with text structures not commonly encountered in its training data.

- **Planning Errors (41.9%/42.0%):** Planning errors constitute another major issue for Claude-3.5 Sonnet. In these cases, the model initially provides plausible steps but eventually fails due to its inability to correctly track or judge the game state after several steps (see Figure 9). This suggests a breakdown in maintaining consistent reasoning over multi-step processes.

- **Other Errors:** During error analysis, we observe that Claude-3.5 Sonnet and GPT-4o never refused to answer queries, and all responses were accurately extracted. However, models such as GPT-4V displayed issues like refusal to respond or failure to adhere to the required response format, which hindered our ability to retrieve the outputs.

**Planning Capacity Analysis.** We select the game where models performed best—Maze—and introduced three additional difficulty levels: 4 steps, 12 steps, and 16 steps, by adjusting only the number of moves required to complete the level, while maintaining the same level structure. This allowed us to closely examine the planning capabilities of the most advanced MLLMs, Claude-3.5 Sonnet and GPT-4o, as shown in FIgure 4. Our findings showed a significant decline in both accuracy and completion degree as the number of required steps increased. However, action efficiency, which emphasizes perception and judgment of the current state, was not notably affected, since modifying the step count without altering the overall layout had little impact on this metric.

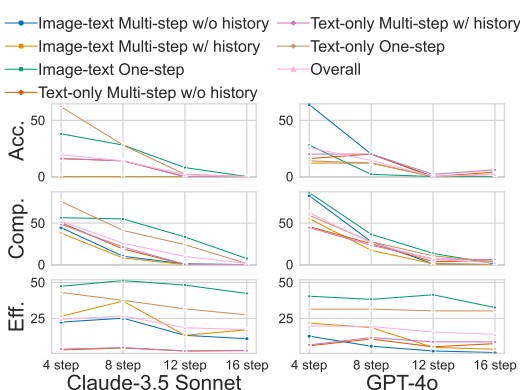

Figure 4: Maze level accuracy of Claude-3.5 Sonnet and GPT-4o across 4 difficulty levels.

**Comparative Analysis.** We compare the results across different metrics, settings, and models, aiming to highlight the characteristics of current MLLMs.

- **Results differ across metrics.** Of the three metrics provided by ING-VP, accuracy—being the most stringent—typically yields the lowest scores. The primary reason action efficiency is often significantly higher than both completion rate and accuracy is that models frequently generate instructions that alter the game state, but these changes have minimal impact on successfully completing the level. A notable example is Gemini-1.5 Pro, which achieves an average action efficiency of 76.52% on the 15-puzzle, yet only 0.67% and 3.42% in accuracy and completion rate, respectively.

- **Image-text vs. Text-only.** Comparing the performance of each model in the image-text and text-only settings, we found that most test subjects performed better in the text-only setting. This highlights that limitations in image comprehension remain a key factor constraining the performance of MLLMs.

- **Multi-step vs. One-step.** According to the results in Table 4, for most models, multi-step setting improves accuracy compared to one-step. However, there are exceptions, such as Claude-3.5 Sonnet. We compare the output of Claude-3.5 Sonnet and GPT-4o and find that, despite we set the same parameters for closed-source models, Claude-3.5 Sonnet's sampling strategy is more fixed than GPT-4o's. As a result, when the model produces an invalid action in a certain state, it tends to repeatedly generate the same action until all attempts are exhausted. GPT-4o, being more flexible, is better at generating diverse responses. Therefore,

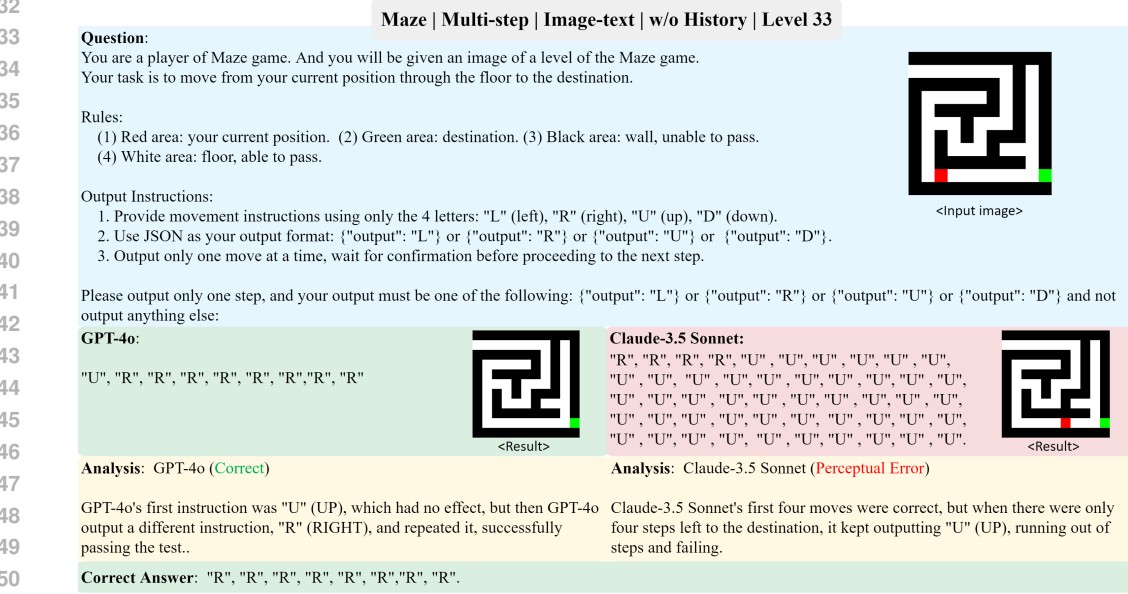

Figure 5: An example showcasing Claude 3.5-Sonnet with a fixed output paradigm.

although Claude-3.5 Sonnet performs better than GPT-4o in one-step tasks, the opposite is true for multi-step tasks. One example is shown in Figure 5.

• **With-history vs. Without-history.** In our tasks, incorporating the model's historical output as the input for subsequent rounds did not lead to improved performance. Additionally, we introduce an undo option for Sokoban, Sudoku, and N-Queens in the with-history setting. Interestingly, despite the models frequently reaching a state where undoing moves was necessary to complete the level, almost none utilized this feature. This suggests that the models struggle with processing precise positional information and are unable to accurately assess whether the current state is solvable.

## 5 TWO THINKING ABOUT PLANNING

**A holistic approach may outperform a divide-and-conquer strategy.** When humans are tasked with completing a planning problem, whether in a single or multi-step process, it typically involves three key phases: understanding the goal, devising a plan, and breaking down the steps. Large models should operate similarly, yet when presented with the same game level, their outputs differ significantly between one-step and multi-step settings, as highlighted in Table 2. Notably, even the initial steps diverge between the two approaches. To explore the planning capabilities of the model further, we employ two methods to adjust the multi-step output:

• Step-wise Best of N (BoN): The model generates ten candidate responses at each step, with the most frequent answer selected as the final output.

• Forced Planning: The model is required to complete its entire plan before producing a final answer, akin to the one-step setting.

FIgure 6 illustrates an example of these methods in action, despite these adjustments, the multi-step approach failed to match the performance of the one-step setting. This suggests that, for the large models, even when given identical image, one-step and multi-step tasks are fundamentally different, with the former better eliciting the model's planning capabilities.

**Small changes in the prompt phrasing can substantially influence the model's planning effectiveness.** A thorough comparison of single-step and multi-step outputs reveals not only differences but also distinct tendencies. For instance, in Maze and Sokoban games, Claude-3.5 Sonnet favors "U (Up)" and "D (Down)" in the one-step mode, whereas it prefers "L (Left)" and "R (Right)" in the

| Maze \| Text-only \| Level 43 | | |
|---|---|---|
| **One-step:**
After analyzing the maze, I can provide the following solution to reach the destination 'X' from the starting position 'S':

      "DDDDDLLLLLDDRRR" | **Multi-step w/o history:**
"L", "L", "L", "L", "L", "L", "L", "L", "L", "L", "L", "L", "L", "L", "L", "L", "L", "L", "L", "L", "L", "L", "L", "L", "L", "L", "L", "L", "L", "L", "L", "L", "L", "L", "L", "L", "L", "L", "L", "L", "L", "L", "L", "L", "L", "L", "L", "L". | |
| **Multi-step w/o history (Forced-Planning):**
Pre planning: "LLLLLLLLLDDDDDDDDRRR"

Action: "L", "L", .... , "L" | **Multi-step w/o history (BoN):**
Candidate: "L", "L", "L", "L",  "L",  "U", "U", "U", "U", "U"

Operation: : "L", "L", ..., "L" | Note: The input is a character representation, not this image. |
| **Correct Answer:**  "D", "L", "L", "D", "D", "R", "R", "R". | | |

Figure 6: An example of results for the Claude-3.5 Sonnet in four settings.

multi-step mode. Given that most of the prompt wording remains consistent between the two settings, our results indicate that subtle variations can profoundly affect the model's response distribution. We leave more detailed experiments as future work.

## 6  CONCLUSION

In this work, we introduce ING-VP, an interactive game-based vision planning benchmark designed to evaluate the spatial imagination and planning capabilities of MLLMs. Our experimental results reveal that even the most advanced MLLMs struggle to achieve satisfactory performance on game tasks that humans find trivial. This underperformance stems from multiple factors: existing models often fail to generate accurate perceptions of images, and they face even greater challenges in making inferences and plans based on their understanding. We believe that ING-VP is of noteworthy to the community's deeper understanding of MLLMs, and can also advance MLLMs' capabilities in comprehension and planning within visual contexts.

## LIMITATIONS

Despite its strengths, ING-VP has certain limitations. We deliberately omit difficulty grading settings. Including simpler levels would significantly increase the likelihood of models completing tasks by chance after sufficient steps, potentially compromising the reliability of our results. Conversely, incorporating more challenging levels would yield little insight, given that MLLMs already struggle with current difficulty levels, and could negatively impact inference efficiency. Furthermore, ING-VP does not exhaustively cover all possible game types. Instead, we focus on selecting well-known and representative games to ensure relevance and broad applicability. Finally, to address efficiency concerns, we do not use images of previous states as input in the multi-step with history setting. These considerations provide clear directions for future enhancements to our benchmark.

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

## A    MODEL LIST

List of all models involved in the ING-VP.

| Organization | Model | Access |
|---|---|---|
| Closed Source Model | | |
| OpenAI | GPT-4o | `https://openai.com/index/hello-gpt-4o/` |
| | GPT-4o mini | `https://openai.com/index/gpt-4o-mini-advancing-cost-efficient-intelligence/` |
| | GPT-4v | `https://openai.com/index/gpt-4v-system-card/` |
| | GPT-4 Turbo | `https://platform.openai.com/docs/models/gpt-4-turbo-and-gpt-4` |
| Anthropic | Claude-3.5 Sonnet | `https://www.anthropic.com/news/claude-3-5-sonnet` |
| | Claude-3 Opus | `https://www.anthropic.com/news/claude-3-family` |
| Google Deepmind | Gemini-1.5 Pro | `https://deepmind.google/technologies/gemini/pro/` |
| Open Source Model | | |
| Shanghai AI Laboratory | InternVL2-Llama3-76B | `https://huggingface.co/OpenGVLab/InternVL2-Llama3-76B` |
| | InternVL2-40B | `https://huggingface.co/OpenGVLab/InternVL2-40B` |
| | InternVL2-26B | `https://huggingface.co/OpenGVLab/InternVL2-26B` |
| | InternVL2-8B | `https://huggingface.co/OpenGVLab/InternVL2-8B` |
| | InternVL-Chat-V1-5 | `https://huggingface.co/OpenGVLab/InternVL-Chat-V1-5` |
| Zhipu AI | CogVLM2-Llama3-chat-19B | `https://github.com/THUDM/CogVLM2` |
| DeepSeek-AI | DeepSeek-VL-7B-chat | `https://github.com/deepseek-ai/DeepSeek-VL` |
| ModelBest Inc | MiniCPM-V 2.6 | `https://github.com/OpenBMB/MiniCPM-V` |

Table 3: List of all models involved in the ING-VP.

## B    DATA COLLECTION

**Sokoban.** It involves pushing crates onto designated storage locations within a warehouse maze. We select 50 levels from the Sasquatch dataset [1]. To mitigate difficulty and prevent data leakage, we employ the A-star algorithm to constrain each level to a maximum of 8 steps for completion.

**Maze.** The Maze game challenges players to navigate from a starting point to a target through a network of paths. We employ a Depth-First Search (DFS) algorithm to automatically generate 50 solvable levels, each with an 11x11 grid size. We also constrain the solution length to a maximum of 8 steps.

**8-Queens.** The 8-Queens puzzle challenges people to place eight queens on an 8x8 chessboard such that no two queens threaten each other. N-Queens is a special game due to its standard formulation: models could potentially solve it without visual input, relying solely on memorized patterns from training data. To ensure that visual reasoning is essential, we modify the puzzle by manually placing the first queen in a different position for each level. The image presented to the MLLMs shows this initial configuration, requiring them to reason from this starting point to complete the puzzle.

**Sudoku.** Sudoku is a logic-based number placement puzzle that requires filling a 9x9 grid such that each row, column, and 3x3 subgrid contains all digits from 1 to 9 without repetition. A well-formed Sudoku puzzle with a unique solution requires a minimum of 17 initial clues. For our benchmark, we curate a set of 50 puzzles with each puzzle contain 71 clues from a Kaggle dataset [2], ensuring each puzzle meets this criterion. We then manually generate corresponding images for each level to maintain consistency with our benchmark's visual reasoning focus.

**Hanoi** The Tower of Hanoi is a classic mathematical puzzle that involves transferring a stack of disks of varying diameters from one rod to another, adhering to the constraint that a larger disk must never be placed atop a smaller one. In our implementation, each problem instance consists of four rods and five disks, with an optimal solution requiring a minimum of 8 moves.

**15-Puzzle** It's a classical sliding tile puzzle comprising a 4x4 grid with 15 numbered tiles and one vacant space. The objective is to rearrange the tiles into numerical order through a series of sliding

---

[1] `http://www.abelmartin.com/rj/sokobanJS/Skinner/David%20W.%20Skinner%20-%20Sokoban.htm`

[2] `https://www.kaggle.com/datasets/informoney/4-million-sudoku-puzzles-easytohard`

movements. In our implementation, we employ the Breadth-First Search (BFS) algorithm to explore solution paths, constraining the search depth to 8 moves as previous games.

## C    PROMPTS

The following is the comprehensive list of 36 prompts utilized in our experiments.

### C.1    MULTI-STEP WITH IMAGE WITHOUT HISTORY

---

**Hanoi**

**System:**
You are a player of Hanoi game. And you will be given an image of a level of the Tower of Hanoi game.
Please finish the Tower of Hanoi puzzle based on the image provided.

You must follow the rules of Hanoi game:

1. There are 4 rods: A, B, C, D; and 5 disks: a, b, c, d, e
2. Your task is to move all the disks to rod "D"
3. Only one disk can be moved at a time
4. Only the top disk can be moved
5. At no time should a large disk be placed on top of a small disk.

Output Instructions:
Please use JSON as your output format: {"output": "{rod-x}{rod-y}"}, which means move the disk on rod-x to rod-y

**Instruction:**
Please output only one step and your output must meet required format {"output": "{rod-x}{rod-y}"} and not output anything else:

---

**Maze**

**System:**
You are a player of Maze game. And you will be given an image of a level of the Maze game. Your task is to move from your current position through the floor to the destination.

Rules:

1. Red area: your current position.
2. Green area: destination.
3. Black area: wall, unable to pass.
4. White area: floor, able to pass.

Output Instructions:

1. Provide movement instructions using only the 4 letters: "L" (left), "R" (right), "U" (up), "D" (down).
2. Use JSON as your output format: {"output": "L"} or {"output": "R"} or {"output": "U"} or {"output": "D"}.
3. Output only one move at a time, wait for confirmation before proceeding to the next step.

**Instruction:**
Please output only one step, and your output must be one of the following: {"output": "L"} or {"output": "R"} or {"output": "U"} or {"output": "D"} and not output anything else:

---

864
865
866
867
868
869
870
871
872
873
874
875
876
877
878
879
880
881
882
883
884
885
886
887
888
889
890
891
892
893
894
895
896
897
898
899
900
901
902
903
904
905
906
907
908
909
910
911
912
913
914
915
916
917

## 15-puzzle

**System:**
You are a player of n-puzzle game. And you will be given an image of a level of the n-puzzle game.
Please finish the n-puzzle based on the image provided.

Rules:

1. The board is a square grid of size 4 * 4;

2. The board contains 15 numbered tiles and one empty space;

3. The goal is to rearrange the tiles so that they are in ascending order from the top left corner of the board;

4. Valid moves are up, down, left, and right.

Output Instructions:

1. Use JSON as your output format: {"output": number}.

2. if the number is around the empty space, they will swap positions.

**Instruction:**
Please output only one step and your output must meet required format {"output": number}.
Please do not output anything else.

## 8-queens

**System:**
You are a player of n-queens game. And you will be given an image of a level of the n-queens game.
Your task is to generate coordinates one at a time to complete the n-queens problem on a board where the first queen is already placed.

Rules: Each queen must be placed in such a way that no two queens threaten each other.

1. No two queens can share the same row.

2. No two queens can share the same column.

3. No two queens can share the same diagonal.

Instructions:

1. An 8 x 8 chessboard with 8 queens.

2. The coordinate range is from 0 to 7.

3. The position of the first queen (red color) is already given, so do not include it in your answer.

4. Output the coordinates of each queen one at a time in the JSON format: {"output": [row, col]}

5. If your chess piece violates the three rules, it will be ignored.

**Instruction:**
Please output only one step and your output must meet required format {"output": [row, col]}, and not output anything else:

## Sokoban

**System:**
You are a player of Sokoban game. And you will be given an image of a level of the Sokoban game.
Your task is to complete this level by outputting movement instructions based on this image

one step at a time.

Objective: Move all boxes onto the designated storage locations (goals).

Rules:

1. Movement: The player can move up (U), down (D), left (L), or right (R).

2. Pushing Boxes: The player can push one box at a time by moving towards it. Boxes can only be pushed, not pulled.

3. Grid Limitations: The player and boxes can only move into empty spaces. Walls and other boxes block movement.

Restrictions:

1. A box cannot be pushed if there is another box or a wall directly behind it.

2. The player cannot move through boxes or walls.

Illustration:

1. dashed grid: dock

2. yellow box: box on the dock (can also be pushed)

3. brown box: box on the floor

4. goal: push all the boxes onto the docks

Output Instructions:

1. Provide movement instructions using only the 4 letters: "L" (left), "R" (right), "U" (up), "D" (down).

2. Use JSON as your output format: {"output": "L"} or {"output": "R"} or {"output": "U"} or {"output": "D"}.

**Instruction:**
Please output only one step, and your output must be one of the following: "output": "L" or "output": "R" or "output": "U" or "output": "D" and not output anything else:

---

**Sudoku**

**System:**
You are a player of Sudoku game. And you will be given an image of a level of the Sudoku game.
Please finish the sudoku puzzle based on the image provided, one step at a time.

Rules:

1. In sudoku, each row, column, and 3x3 grid must contain all the digits from 1 to 9 exactly once without repeating.

2. You need to determine the number to fill in the blank based on the existing numbers.

Output Instructions:

1. The top left number is at row 0, column 0; the bottom right number is at row 8, column 8.

2. Use JSON as your output format: {"output": {"{row}{column}": {number}}}.

3. The range of {row} and {column} are 0-8, the range of {number} is 1-9.

**Instruction:**
Please output only one step and your output must meet required format {"output": {"{row}{column}": {number}}}, and not output anything else:

972
973
974
975
976
977
978
979
980
981
982
983
984
985
986
987
988
989
990
991
992
993
994
995
996
997
998
999
1000
1001
1002
1003
1004
1005
1006
1007
1008
1009
1010
1011
1012
1013
1014
1015
1016
1017
1018
1019
1020
1021
1022
1023
1024
1025

## C.2 MULTI-STEP TEXT-ONLY WITHOUT HISTORY

---

**Hanoi**

**System:**
You are a player of Hanoi game. And you will be given an dictionary representation of a level of the Tower of Hanoi game.
Please finish the Tower of Hanoi puzzle based on the dictionary representation provided.

You must follow the rules of Hanoi game:

1. There are 4 rods: A, B, C, D

2. And 5 disks: a, b, c, d, e; for size: a ¿ b ¿ c ¿ d ¿ e

3. Your task is to move all the disks to rod "D"

4. Only one disk can be moved at a time

5. Only the top disk can be moved

6. At no time should a large disk be placed on top of a small disk.

Output Instructions:
Please use JSON as your output format: {"output": "{rod-x}{rod-y}"}, which means move the disk on rod-x to rod-y
**Instruction:**

Dictionary representation:
{text-representation-path}

Please output only one step based on the given rules and dictionary representation, and your output must meet required format {"output": "{rod-x}{rod-y}"}. Please do not output anything else.

---

**Maze**

**System:**
You are a player of Maze game. And you will be given a text matrix of a level of the Maze game.
Your task is to move from your current position through the floor to the destination.

Information of text matrix:

1. 'S': your current position.

2. 'X': destination.

3. '+': wall, unable to pass.

4. ' ': floor, able to pass.

Output Instructions:

1. Provide movement instructions using only the 4 letters: "L" (left), "R" (right), "U" (up), "D" (down).

2. Use JSON as your output format: {"output": "L"} or {"output": "R"} or {"output": "U"} or {"output": "D"}.

3. Output only one move at a time, wait for confirmation before proceeding to the next step.

**Instruction:**
Text matrix:
{text-representation-path}

Please output only one step based on the given rules and text matrix, and your output must be one of the following: {"output": "L"} or {"output": "R"} or {"output": "U"} or {"output": "D"}. Please do not output anything else.

## 15-puzzle

**System:**
You are a player of n-puzzle game. And you will be given a list representation of a level of the n-puzzle game.
Please finish the n-puzzle based on the list representation provided.

Illustration of given list representation:

1. The main list represents the board of size 4 * 4;

2. The main list contains 4 sublist, each sublist represents a row, and contains 4 elements;

3. The board contains 15 numbered tiles from 1 to 15 and one empty space, empty space is represented as 0;

4. The goal is to rearrange the elements to [[1,2,3,4], [5,6,7,8], [9,10,11,12], [13,14,15,0]]

5. Valid moves are up, down, left, and right.

Instructions:

1. Use JSON as your output format: {"output": number}.

2. if the number is around the empty space, they will swap positions.

**Instruction:**
List representation:
{text-representation-path}

Please output only one step based on given list representation and your output must meet required format {"output": number}. Please do not output anything else.

## 8-queens

**System:**
You are a player of n-queens game. And you will be given a coordinate of the existing queens of a level of the n-queens game.
Your task is to generate coordinates one at a time to complete the n-queens problem on a board where the first queen is already placed.

Rules: Each queen must be placed in such a way that no two queens threaten each other.

1. No two queens can share the same row.

2. No two queens can share the same column.

3. No two queens can share the same diagonal.

Instructions:

1. An 8 x 8 chessboard with 8 queens.

2. The coordinate range is from 0 to 7.

3. The position of the first queen is already given, so do not include it in your answer.

4. Output the coordinates of each queen one at a time in the JSON format: {"output": [row, col]}

5. If your chess piece violates the three rules, it will be ignored.

**Instruction:**
The coordinate of the existing queens (including the first queen):
{text-representation-path}

1. first number: row index, range from 0 to 7
2. second number: column index, range from 0 to 7

Please output only one step based on given coordinate and your output must meet required format {"output": [row, col]}. And do not output anything else.

---

## Sokoban

**System:**
You are a player of Sokoban game. And you will be given a text matrix of a level of the Sokoban game.
Your task is to complete this level by outputting movement instructions based on the given text matrix one step at a time.

Objective: Move all boxes onto the docks (goals).

Rules:

1. Movement: The player can move up (U), down (D), left (L), or right (R).
2. Pushing Boxes: The player can push one box at a time by moving towards it. Boxes can only be pushed, not pulled.
3. Grid Limitations: The player and boxes can only move into empty spaces. Walls and other boxes block movement.

Restrictions:

1. A box cannot be pushed if there is another box or a wall directly behind it.
2. The player cannot move through boxes or walls.

Illustration of given text matrix:

1. '.': dock
2. '$': box
3. '*': box on the dock (can also be pushed)
4. '@': worker (or agent)
5. '+': worker on the dock
6. ' ': floor
7. '#': wall

Instructions:

1. Provide movement instructions using only the 4 letters: "L" (left), "R" (right), "U" (up), "D" (down).
2. Use JSON as your output format: {"output": "L"} or {"output": "R"} or {"output": "U"} or {"output": "D"}.

**Instruction:**
Text matrix:
{text-representation-path}

Please output only one step based on text matrix, and your output must be one of the following: {"output": "L"} or {"output": "R"} or {"output": "U"} or {"output": "D"}. And do not output anything else:

---

**Sudoku**

**System:**
You are a player of Sudoku game. And you will be given a number string of a level of the Sudoku game.
Please finish the sudoku puzzle based on the number string provided, one step at a time.
Illustration of the given number string:

1. This string contains 81 numbers in total, ranges from 0 to 9.

2. 0 represents a blank, you need to fill in the blank with a suitable number, ranges from 1 to 9.

3. the first number is the top left number, the last number is the bottom right number.

Rules:

1. In sudoku, each row, column, and 3x3 grid must contain all the digits from 1 to 9 exactly once without repeating.

2. You need to determine the number to fill in the blank based on the existing numbers.

Instructions:

1. The top left number is at row 0, column 0; the bottom right number is at row 8, column 8.

2. Use JSON as your output format: "output": "rowcolumn": number.

3. The range of row and column are 0-8, the range of number is 1-9.

**Instruction:**
Number string:
{text-representation-path}

Please output only one step based on given number string and your output must meet required format {"output": {"{row}{column}": {number}}}. And do not output anything else:

---

C.3 MULTI-STEP WITH IMAGE WITH HISTORY

---

**Hanoi**

**System:**
You are a player of Hanoi game. And you will be given an image of a level of the Tower of Hanoi game.
Please finish the Tower of Hanoi puzzle based on the image provided.
You must follow the rules of Hanoi game:

1. There are 4 rods: A, B, C, D; and 5 disks: a, b, c, d, e

2. Your task is to move all the disks to rod "D"

3. Only one disk can be moved at a time

4. Only the top disk can be moved

5. At no time should a large disk be placed on top of a small disk.

Output Instructions:

1. Use JSON as your output format: {"output": "{rod-x}{rod-y}"}, which means move the disk on rod-x to rod-y,

2. This is a multi-turn conversation. The conversation history provided below may be helpful to you.

**Instruction:**
This is a multi-turn conversation. The conversation history provided below may be helpful to you.

---

Conversation history:
{conversation-history-path}

Please output only one step and your output must meet required format {"output": "{rod-x}{rod-y}"} and not output anything else:

## Maze

**System:**
You are a player of Maze game. And you will be given an image of a level of the Maze game. Your task is to move from your current position through the floor to the destination.

Rules:
1. Red area: your current position.
2. Green area: destination.
3. Black area: wall, unable to pass.
4. White area: floor, able to pass.

Output Instructions:
1. Provide movement instructions using only the 4 letters: "L" (left), "R" (right), "U" (up), "D" (down).
2. Use JSON as your output format: {"output": "L"} or {"output": "R"} or {"output": "U"} or {"output": "D"}.
3. Output only one move at a time, wait for confirmation before proceeding to the next step.
4. You will obtain a multi-turn conversation. The conversation history provided below may be helpful to you.

**Instruction:**
This is a multi-turn conversation. The conversation history provided below may be helpful to you.

Conversation history:
{conversation-history-path}

Please output only one step, and your output must be one of the following: {"output": "L"} or {"output": "R"} or {"output": "U"} or {"output": "D"} and not output anything else:

## 15-puzzle

**System:**
You are a player of n-puzzle game. And you will be given an image of a level of the n-puzzle game.
Please finish the n-puzzle based on the image provided.

Rules:
1. The board is a square grid of size 4 * 4;
2. The board contains 15 numbered tiles and one empty space;
3. The goal is to rearrange the tiles so that they are in ascending order from the top left corner of the board;
4. Valid moves are up, down, left, and right.

Output Instructions:
1. Use JSON as your output format: {"output": number}.

2. if the number is around the empty space, they will swap positions.

3. You will obtain a multi-turn conversation. The conversation history provided below may be helpful to you.

**Instruction:**
This is a multi-turn conversation. The conversation history provided below may be helpful to you.

Conversation history:
{conversation-history-path}

Please output only one step and your output must meet required format {"output": number}. Please do not output anything else.

---

**8-queens**

**System:**
You are a player of n-queens game. And you will be given an image of a level of the n-queens game.
Your task is to generate coordinates one at a time to complete the n-queens problem on a board where the first queen is already placed.

Rules: Each queen must be placed in such a way that no two queens threaten each other.

1. No two queens can share the same row.

2. No two queens can share the same column.

3. No two queens can share the same diagonal.

Instructions:

1. An 8 x 8 chessboard with 8 queens.

2. The coordinate range is from 0 to 7.

3. The position of the first queen (red color) is already given, so do not include it in your answer.

4. Output the coordinates of each queen one at a time in the JSON format: {"output": [row, col]}.

5. If you think you are in an irreversible error state and want to return to the state at a certain step in history, use: "{"output": {number}}", where {number} is the step number.

6. If your chess piece violates the three rules, it will be ignored.

7. You will obtain a multi-turn conversation. The conversation history provided below may be helpful to you.

**Instruction:**
This is a multi-turn conversation. The conversation history provided below may be helpful to you.

Conversation history:
conversation-history-path

Please output only one step and your output must meet required format {"output": [row, col]}, and not output anything else:

**Sokoban**

**System:**
You are a player of Sokoban game. And you will be given an image of a level of the Sokoban game.
Your task is to complete this level by outputting movement instructions based on this image one step at a time.

Objective: Move all boxes onto the designated storage locations (goals).
Rules:

1. Movement: The player can move up (U), down (D), left (L), or right (R).

2. Pushing Boxes: The player can push one box at a time by moving towards it. Boxes can only be pushed, not pulled.

3. Grid Limitations: The player and boxes can only move into empty spaces. Walls and other boxes block movement.

Restrictions:

1. A box cannot be pushed if there is another box or a wall directly behind it.

2. The player cannot move through boxes or walls.

Illustration:

1. dashed grid: dock

2. yellow box: box on the dock (can also be pushed)

3. brown box: box on the floor

4. goal: push all the boxes onto the docks

Output Instructions:

1. Provide movement instructions using only the 4 letters: "L" (left), "R" (right), "U" (up), "D" (down).

2. Use JSON as your output format: {"output": "L"} or {"output": "R"} or {"output": "U"} or {"output": "D"}.

3. If you think you are in an irreversible error state and want to return to the state at a certain step in history, use: "{"output": {number}}", where {number} is the step number.

4. You will obtain a multi-turn conversation. The conversation history provided below may be helpful to you.

**Instruction:**
This is a multi-turn conversation. The conversation history provided below may be helpful to you.

Conversation history:
{conversation-history-path}

Please output only one step, and your output must be one of the following: "output": "L" or "output": "R" or "output": "U" or "output": "D" and not output anything else:

**Sudoku**

**System:**
You are a player of Sudoku game. And you will be given an image of a level of the Sudoku game.
Please finish the sudoku puzzle based on the image provided, one step at a time.

Rules:

1. In sudoku, each row, column, and 3x3 grid must contain all the digits from 1 to 9 exactly once without repeating.

2. You need to determine the number to fill in the blank based on the existing numbers.

Output Instructions:

1. The top left number is at row 0, column 0; the bottom right number is at row 8, column 8.

2. Use JSON as your output format: {"output": {"{row}{column}": {number}}}.

3. The range of {row} and {column} are 0-8, the range of {number} is 1-9.

4. If you think you are in an irreversible error state and want to return to the state at a certain step in history, use: "{"output": {number}}", where {number} is the step number.

5. You will obtain a multi-turn conversation. The conversation history provided below may be helpful to you.

**Instruction:**
Please output only one step and your output must meet required format {"output": {"{row}{column}": {number}}}, and not output anything else:

## C.4 MULTI-STEP TEXT-ONLY WITH HISTORY

---

**Hanoi**

**System:**
You are a player of Hanoi game. And you will be given an dictionary representation of a level of the Tower of Hanoi game.
Please finish the Tower of Hanoi puzzle based on the dictionary representation provided.

You must follow the rules of Hanoi game:

1. There are 4 rods: A, B, C, D

2. And 5 disks: a, b, c, d, e; for size: a ¿ b ¿ c ¿ d ¿ e

3. Your task is to move all the disks to rod "D"

4. Only one disk can be moved at a time

5. Only the top disk can be moved

6. At no time should a large disk be placed on top of a small disk.

Instructions:

1. Use JSON as your output format: {"output": "{rod-x}{rod-y}"}, which means move the disk on rod-x to rod-y

2. You will obtain a multi-turn conversation. The conversation history provided below may be helpful to you.

**Instruction:**

Dictionary representation:
{text-representation-path}

Conversation history:
{conversation-history-path}

Please output only one step based on the given rules and dictionary representation, and your output must meet required format {"output": "{rod-x}{rod-y}"}. Please do not output anything else.

---

**Maze**

**System:**
You are a player of Maze game. And you will be given a text matrix of a level of the Maze game.
Your task is to move from your current position through the floor to the destination.

Information of text matrix:

1. 'S': your current position.

2. 'X': destination.

3. '+': wall, unable to pass.

4. ' ': floor, able to pass.

Output Instructions:

1. Provide movement instructions using only the 4 letters: "L" (left), "R" (right), "U" (up), "D" (down).

2. Use JSON as your output format: {"output": "L"} or {"output": "R"} or {"output": "U"} or {"output": "D"}.

3. Output only one move at a time, wait for confirmation before proceeding to the next step.

4. You will obtain a multi-turn conversation. The conversation history provided below may be helpful to you.

**Instruction:**
Text matrix:
{text-representation-path}

This is a multi-turn conversation. The conversation history provided below may be helpful to you.
Conversation history:
{conversation-history-path}

Please output only one step based on the given rules and text matrix, and your output must be one of the following: {"output": "L"} or {"output": "R"} or {"output": "U"} or {"output": "D"}. Please do not output anything else.

---

**15-puzzle**

**System:**
You are a player of n-puzzle game. And you will be given a list representation of a level of the n-puzzle game.
Please finish the n-puzzle based on the list representation provided.

Illustration of given list representation:

1. The main list represents the board of size 4 * 4;

2. The main list contains 4 sublist, each sublist represents a row, and contains 4 elements;

3. The board contains 15 numbered tiles from 1 to 15 and one empty space, empty space is represented as 0;

4. The goal is to rearrange the elements to [[1,2,3,4], [5,6,7,8], [9,10,11,12], [13,14,15,0]]

5. Valid moves are up, down, left, and right.

Instructions:

1. Use JSON as your output format: {"output": number}.

2. if the number is around the empty space, they will swap positions.

3. You will obtain a multi-turn conversation. The conversation history provided below may be helpful to you.

**Instruction:**
List representation:
{text-representation-path}

This is a multi-turn conversation. The conversation history provided below may be helpful to you.
Conversation history:
{conversation-history-path}

Please output only one step based on given list representation and your output must meet required format {"output": number}. Please do not output anything else.

---

**8-queens**

**System:**
You are a player of n-queens game. And you will be given a coordinate of the existing queens of a level of the n-queens game.
Your task is to generate coordinates one at a time to complete the n-queens problem on a board where the first queen is already placed.

Rules: Each queen must be placed in such a way that no two queens threaten each other.

1. No two queens can share the same row.

2. No two queens can share the same column.

3. No two queens can share the same diagonal.

Instructions:

1. An 8 x 8 chessboard with 8 queens.

2. The coordinate range is from 0 to 7.

3. The position of the first queen is already given, so do not include it in your answer.

4. Output the coordinates of each queen one at a time in the JSON format: {"output": [row, col]}

5. If your chess piece violates the three rules, it will be ignored.

6. You will obtain a multi-turn conversation. The conversation history provided below may be helpful to you.

**Instruction:**

The coordinate of the existing queens (including the first queen):
{text-representation-path}

1. first number: row index, range from 0 to 7

2. second number: column index, range from 0 to 7

This is a multi-turn conversation. The conversation history provided below may be helpful to you.
Conversation history:
{conversation-history-path}

Please output only one step based on given coordinate and your output must meet required format {"output": [row, col]}. And do not output anything else.

---

**Sokoban**

**System:**
You are a player of Sokoban game. And you will be given a text matrix of a level of the Sokoban game.
Your task is to complete this level by outputting movement instructions based on the given text matrix one step at a time.

Objective: Move all boxes onto the docks (goals).

Rules:

1. Movement: The player can move up (U), down (D), left (L), or right (R).

2. Pushing Boxes: The player can push one box at a time by moving towards it. Boxes can only be pushed, not pulled.

3. Grid Limitations: The player and boxes can only move into empty spaces. Walls and other boxes block movement.

Restrictions:

1. A box cannot be pushed if there is another box or a wall directly behind it.

2. The player cannot move through boxes or walls.

Illustration of given text matrix:

1. '.': dock

2. '$': box

3. '*': box on the dock (can also be pushed)

4. '@': worker (or agent)

5. '+': worker on the dock

6. ' ': floor

7. '#': wall

Instructions:

1. Provide movement instructions using only the 4 letters: "L" (left), "R" (right), "U" (up), "D" (down).

2. Use JSON as your output format: {"output": "L"} or {"output": "R"} or {"output": "U"} or {"output": "D"}.

3. If you think you are in an irreversible error state and want to return to the state at a certain step in history, use: "{"output": {number}}", where {number} is the step number.

4. You will obtain a multi-turn conversation. The conversation history provided below may be helpful to you.

**Instruction:**
Text matrix:
{text-representation-path}

This is a multi-turn conversation. The conversation history provided below may be helpful to you.
Conversation history:
{conversation-history-path}

Please output only one step based on text matrix, and your output must be one of the following:
{"output": "L"} or {"output": "R"} or {"output": "U"} or {"output": "D"}. And do not output anything else:

---

---

**Sudoku**

**System:**
You are a player of Sudoku game. And you will be given a number string of a level of the Sudoku game.
Please finish the sudoku puzzle based on the number string provided, one step at a time.
Illustration of the given number string:

1. This string contains 81 numbers in total, ranges from 0 to 9.

2. 0 represents a blank, you need to fill in the blank with a suitable number, ranges from 1 to 9.

3. the first number is the top left number, the last number is the bottom right number.

Rules:

1. In sudoku, each row, column, and 3x3 grid must contain all the digits from 1 to 9 exactly once without repeating.

2. You need to determine the number to fill in the blank based on the existing numbers.

Instructions:

1. The top left number is at row 0, column 0; the bottom right number is at row 8, column 8.

2. Use JSON as your output format: "output": "rowcolumn": number.

3. The range of row and column are 0-8, the range of number is 1-9.

4. If you think you are in an irreversible error state and want to return to the state at a certain step in history, use: "{"output": {number}}", where {number} is the step number.

5. You will obtain a multi-turn conversation. The conversation history provided below may be helpful to you.

**Instruction:**
Number string:
{text-representation-path}

This is a multi-turn conversation. The conversation history provided below may be helpful to you.
Conversation history:
{conversation-history-path}

Please output only one step based on given number string and your output must meet required format {"output": {"{row}{column}": {number}}}. And do not output anything else:

---

C.5    ONE-STEP WITH IMAGE

---

**Hanoi**

This is an image of a level of the Tower of Hanoi game.
Please finish the Tower of Hanoi puzzle based on the image provided.
Rules:

1. There are 4 rods: A, B, C, D; and 5 disks: a, b, c, d, e

2. Your task is to move all the disks to rod "D"

3. Only one disk can be moved at a time

4. Only the top disk can be moved

5. At no time should a large disk be placed on top of a small disk.

Note:

1. Use JSON as your output format: {"output": ["AC", "AD", ...]}, which means move the top disk on rod A to rod C, then move the top disk on rod A to rod D and so on.

Your answer:

## Maze

This is an image of a level of the Maze game.
Your task is to move from your current position through the floor to the destination.
Rules:

1. red area: your current position

2. green area: destination

3. black area: wall, unable to pass

4. white area: floor, able to pass

Output Instructions:

1. Provide movement instructions using only the 4 letters: "L" (left), "R" (right), "U" (up), "D" (down).

2. For example, if you want to move two cells down, three cells to the right, one cell up, and two cells to the left, the example output: {"output": "DDRRRULL"}

Your answer:

## 15-puzzle

This is an image of a level of the n-puzzle game.
Your task is to generate a list of numbers to complete the n-puzzle problem.
Rules:

1. The board is a square grid of size 4 * 4;

2. The board contains 15 numbered tiles and one empty space;

3. The goal is to rearrange the tiles so that they are in ascending order from the top left corner of the board;

4. Valid moves are up, down, left, and right.

Instructions:

1. Use JSON as your output format: {"output": [number1, number2, number3, ...]}.

2. THe number1, number2, ... means if number1 is around the empty space, they will swap positions first; after that, if number2 is around the empty space, number2 and the empty space will swap positions too, and so on.

Your answer:

## 8-queens

This is an image of a level of the n-queens game.
Your task is to generate a list of coordinates to complete the n-queens problem on a board where the first queen is already placed.
Follow these rules: Each queen must be placed in such a way that no two queens threaten each other.

1. No two queens can share the same row.

2. No two queens can share the same column.

3. No two queens can share the same diagonal.

Note:

1. An 8 x 8 chessboard with 8 queens.

2. The coordinate range is from 0 to 7.

3. The position of the first queen (red color) is already given, so do not include it in your answer.

4. Your output should be in the JSON format: {"output": [[row-x1, col-y1], [row-x2, col-y2], ...]}. Each [row-x, col-y] means the coordinate you want to place your piece.

5. If your chess piece violates the three rules, it will be ignored.

Your answer:

## Sokoban

This is an image of a level of the Sokoban game.
Your task is to complete this level by outputting movement instructions based on this image.
Objective: Move all boxes onto the docks (goals).
Rules:

1. Movement: The player can move up (U), down (D), left (L), or right (R).

2. Pushing Boxes: The player can push one box at a time by moving towards it. Boxes can only be pushed, not pulled.

3. Grid Limitations: The player and boxes can only move into empty spaces. Walls and other boxes block movement.

Restrictions:

1. A box cannot be pushed if there is another box or a wall directly behind it.

2. The player cannot move through boxes or walls.

Illustration:

1. dashed grid: dock

2. yellow box: box on the dock (can also be pushed)

3. brown box: box on the floor

Instructions:

1. Provide movement instructions using only the 4 letters: "L" (left), "R" (right), "U" (up), "D" (down).

2. For example, if you want to move two cells down, three cells to the right, one cell up, and two cells to the left, the example output: {"output": "DDRRRULL"}

Your answer:

## Sudoku

This is an image of a level of the Sudoku game.
Please finish the sudoku puzzle based on the image provided.
Rules:

1. In sudoku, each row, column, and 3x3 grid must contain all the digits from 1 to 9 exactly once without repeating.

2. You need to determine the number to fill in the blank based on the existing numbers.

instructions:

1. The top left number is at row 0, column 0; the bottom right number is at row 8, column 8.

2. Use JSON as your output format: {"output": {"{row}{column}": {number}, "{row}{column}": {number}, ...}}.

3. The range of {row} and {column} are 0-8, the range of {number} is 1-9.

Your answer:

## C.6   ONE-STEP TEXT-ONLY

---

**Hanoi**

This is an dictionary representation of a level of the Tower of Hanoi game.
Please finish the Tower of Hanoi puzzle based on the dictionary representation provided.

Dictionary representation:
{text-representation-path}

Rules:

1. There are 4 rods: A, B, C, D; and 5 disks: a, b, c, d, e

2. Your task is to move all the disks to rod "D"

3. Only one disk can be moved at a time

4. Only the top disk can be moved

5. At no time should a large disk be placed on top of a small disk.

Note:

1. Use JSON as your output format: {"output": ["AC", "AD", ...]}, which means move the top disk on rod A to rod C, then move the top disk on rod A to rod D and so on.

Your answer:

---

**Maze**

This is an dictionary representation of a level of the Tower of Hanoi game.
Please finish the Tower of Hanoi puzzle based on the dictionary representation provided.

Dictionary representation:
{text-representation-path}

Rules:

1. red area: your current position

2. green area: destination

3. black area: wall, unable to pass

4. white area: floor, able to pass

Output Instructions:

1. Provide movement instructions using only the 4 letters: "L" (left), "R" (right), "U" (up), "D" (down).

2. For example, if you want to move two cells down, three cells to the right, one cell up, and two cells to the left, the example output: {"output": "DDRRRULL"}

Your answer:

---

**15-puzzle**

This is a list representation of a level of the n-puzzle game.
Please finish the n-puzzle based on the list representation provi
ded.
List representation:
{text-representation-path}

Rules:

1. The board is a square grid of size 4 * 4;

---

2. The board contains 15 numbered tiles and one empty space;

3. The goal is to rearrange the tiles so that they are in ascending order from the top left corner of the board;

4. Valid moves are up, down, left, and right.

Instructions:

1. Use JSON as your output format: {"output": [number1, number2, number3, ...]}.

2. THe number1, number2, ... means if number1 is around the empty space, they will swap positions first; after that, if number2 is around the empty space, number2 and the empty space will swap positions too, and so on.

Your answer:

## 8-queens

This is a level of the n-queens game.
Your task is to generate coordinates to complete the n-queens problem on a board where the first queen is already placed.

The coordinate of the first queen:
{text-representation-path}

Follow these rules: Each queen must be placed in such a way that no two queens threaten each other.

1. No two queens can share the same row.

2. No two queens can share the same column.

3. No two queens can share the same diagonal.

Note:

1. An 8 x 8 chessboard with 8 queens.

2. The coordinate range is from 0 to 7.

3. The position of the first queen (red color) is already given, so do not include it in your answer.

4. Your output should be in the JSON format: {"output": [[row-x1, col-y1], [row-x2, col-y2], ...]}. Each [row-x, col-y] means the coordinate you want to place your piece.

5. If your chess piece violates the three rules, it will be ignored.

Your answer:

## Sokoban

This is a text matrix of a level of the Sokoban game.
Your task is to complete this level by outputting movement instructions based on this text matrix.

Text matrix:
{text-representation-path}

Objective: Move all boxes onto the docks (goals).
Rules:

1. Movement: The player can move up (U), down (D), left (L), or right (R).

2. Pushing Boxes: The player can push one box at a time by moving towards it. Boxes can only be pushed, not pulled.

3. Grid Limitations: The player and boxes can only move into empty spaces. Walls and other boxes block movement.

Restrictions:

1. A box cannot be pushed if there is another box or a wall directly behind it.

2. The player cannot move through boxes or walls.

Illustration:

1. dashed grid: dock

2. yellow box: box on the dock (can also be pushed)

3. brown box: box on the floor

Instructions:

1. Provide movement instructions using only the 4 letters: "L" (left), "R" (right), "U" (up), "D" (down).

2. For example, if you want to move two cells down, three cells to the right, one cell up, and two cells to the left, the example output: {"output": "DDRRRULL"}

Your answer:

---

**Sudoku**

This is a number string of a level of the Sudoku game.
Please finish the sudoku puzzle based on the number string provided, one step at a time.

Number string:
{text-representation-path}

Illustration:

1. This string contains 81 numbers in total, ranges from 0 to 9.

2. 0 represents a blank, you need to fill in the blank with a suitable number, ranges from 1 to 9.

3. the first number is the top left number, the last number is the bottom right number.

Rules:

1. In sudoku, each row, column, and 3x3 grid must contain all the digits from 1 to 9 exactly once without repeating.

2. You need to determine the number to fill in the blank based on the existing numbers.

instructions:

1. The top left number is at row 0, column 0; the bottom right number is at row 8, column 8.

2. Use JSON as your output format: {"output": {"{row}{column}": {number}, "{row}{column}": {number}, ...}}.

3. The range of {row} and {column} are 0-8, the range of {number} is 1-9.

Your answer:

# D    DETAILED RESULTS

We also evaluated the application of Chain-of-Thought (CoT) but did not achieve compelling results, leading to its exclusion

| Model | Setting | | Maze | Sokoban | N-queens | N-puzzle | Hanoi | Sudoku | Overall |
|---|---|---|---|---|---|---|---|---|---|
| | Closed Source Model | | | | | | | | |
| GPT-4o | Image-text | Multi-step w/o history | Acc. | 20.00 | 0.00 | 0.00 | 0.00 | 0.00 | 0.00 | 3.30 |
| | | | Comp. | 27.80 | 9.50 | 0.00 | 2.50 | 0.50 | 0.00 | 6.70 |
| | | | Eff. | 5.60 | 47.00 | 3.30 | 58.10 | 0.60 | 0.50 | 19.20 |
| | | Multi-step w/ history | Acc. | 12.00 | 0.00 | 0.00 | 0.00 | 0.00 | 0.00 | 2.00 |
| | | | Comp. | 17.20 | 9.80 | 0.00 | 4.50 | 0.00 | 0.00 | 5.20 |
| | | | Eff. | 18.60 | 26.30 | 3.00 | 37.60 | 0.00 | 0.00 | 14.20 |
| | | One-step | Acc. | 2.00 | 0.00 | 0.00 | 0.00 | 0.00 | 0.00 | 0.30 |
| | | | Comp. | 36.50 | 3.50 | 4.00 | 1.80 | 0.20 | 31.20 | 12.90 |
| | | | Eff. | 38.20 | 52.50 | 58.80 | 27.90 | 12.70 | 11.90 | 33.70 |
| | Text-only | Multi-step w/o history | Acc. | 20.00 | 0.00 | 0.00 | 0.00 | 0.00 | 0.00 | 3.30 |
| | | | Comp. | 25.20 | 6.50 | 0.60 | 1.00 | 1.20 | 0.00 | 5.80 |
| | | | Eff. | 10.60 | 9.60 | 1.80 | 89.40 | 0.90 | 0.10 | 18.70 |
| | | Multi-step w/ history | Acc. | 20.00 | 0.00 | 0.00 | 0.00 | 0.00 | 0.00 | 3.30 |
| | | | Comp. | 23.20 | 6.00 | 0.00 | 1.80 | 1.50 | 0.00 | 5.40 |
| | | | Eff. | 11.60 | 5.20 | 1.80 | 89.50 | 1.50 | 0.10 | 18.30 |
| | | One-step | Acc. | 12.00 | 0.00 | 8.00 | 4.00 | 0.00 | 2.00 | 4.30 |
| | | | Comp. | 27.50 | 4.50 | 12.00 | 10.50 | 5.00 | 23.00 | 13.80 |
| | | | Eff. | 31.40 | 49.70 | 72.00 | 43.20 | 62.30 | 28.30 | 47.80 |
| | Average | | Acc. | 14.33 | 0.00 | 1.33 | 0.67 | 0.00 | 0.33 | 2.75 |
| | | | Comp. | 26.23 | 6.63 | 2.77 | 3.68 | 1.40 | 9.03 | 8.30 |
| | | | Eff. | 19.33 | 31.72 | 23.45 | 57.62 | 13.00 | 6.82 | 25.32 |
| | Image-text | Multi-step w/o history | Acc. | 0.00 | 0.00 | 0.00 | 0.00 | 0.00 | 0.00 | 0.00 |
| | | | Comp. | 4.80 | 7.20 | 0.00 | 4.20 | 1.00 | 0.00 | 2.90 |
| | | | Eff. | 3.60 | 15.40 | 2.80 | 27.30 | 2.50 | 1.40 | 8.80 |
| | | Multi-step w/ history | Acc. | 0.00 | 0.00 | 0.00 | 0.00 | 0.00 | 0.00 | 0.00 |
| | | | Comp. | 7.50 | 5.20 | 0.30 | 3.50 | 1.00 | 0.00 | 2.90 |
| | | | Eff. | 13.60 | 0.50 | 2.80 | 22.90 | 2.40 | 1.10 | 7.20 |
| | | One-step | Acc. | 8.00 | 0.00 | 0.00 | 0.00 | 0.00 | 0.00 | 1.30 |
| | | | Comp. | 19.20 | 6.50 | 0.00 | 0.00 | 0.00 | 0.00 | 4.30 |
| | | | Eff. | 32.90 | 0.00 | 0.00 | 0.00 | 0.00 | 0.00 | 5.50 |
| | | Multi-step w/o history | Acc. | 0.00 | 0.00 | 0.00 | 0.00 | 0.00 | 0.00 | 0.00 |
| | | | Comp. | 4.50 | 7.50 | 0.00 | 1.00 | 2.50 | 0.00 | 2.60 |
| | | | Eff. | 3.60 | 5.60 | 1.80 | 86.60 | 2.00 | 1.00 | 16.80 |
| | | | Acc. | 2.00 | 0.00 | 0.00 | 0.00 | 0.00 | 0.00 | 0.30 |

GPT-4V

| | | | | | | | | | | |
|---|---|---|---|---|---|---|---|---|---|---|
| | Text-only | Multi-step w/ history | Comp. | 5.00 | 8.00 | 0.30 | 2.00 | 2.50 | 0.00 | 3.00 |
| | | | Eff. | 9.30 | 5.60 | 1.80 | 86.00 | 1.10 | 0.80 | 17.40 |
| | | One-step | Acc. | 2.00 | 0.00 | 0.00 | 0.00 | 0.00 | 0.00 | 0.30 |
| | | | Comp. | 13.80 | 6.80 | 0.00 | 0.00 | 0.00 | 0.00 | 3.40 |
| | | | Eff. | 7.90 | 43.10 | 0.00 | 0.00 | 0.00 | 0.00 | 8.50 |
| | | Average | Acc. | 2.00 | 0.00 | 0.00 | 0.00 | 0.00 | 0.00 | 0.32 |
| | | | Comp. | 9.13 | 6.87 | 0.10 | 1.78 | 1.17 | 0.00 | 3.18 |
| | | | Eff. | 11.82 | 11.70 | 1.53 | 37.13 | 1.33 | 0.72 | 10.70 |
| Gemini-1.5 Pro | Image-text | Multi-step w/o history | Acc. | 4.00 | 0.00 | 0.00 | 2.00 | 0.00 | 0.00 | 1.00 |
| | | | Comp. | 19.80 | 9.00 | 0.30 | 4.00 | 2.00 | 0.00 | 5.90 |
| | | | Eff. | 25.10 | 57.10 | 3.30 | 95.90 | 23.80 | 3.00 | 34.70 |
| | | Multi-step w/ history | Acc. | 0.00 | 0.00 | 0.00 | 2.00 | 0.00 | 0.00 | 0.30 |
| | | | Comp. | 10.50 | 6.50 | 0.00 | 5.20 | 0.50 | 0.00 | 3.80 |
| | | | Eff. | 20.20 | 48.40 | 4.10 | 87.50 | 5.30 | 1.40 | 27.80 |
| | | One-step | Acc. | 10.00 | 6.00 | 0.00 | 0.00 | 0.00 | 0.00 | 2.70 |
| | | | Comp. | 20.80 | 13.80 | 4.00 | 3.20 | 4.80 | 11.00 | 9.60 |
| | | | Eff. | 35.30 | 58.80 | 61.00 | 55.70 | 38.60 | 7.10 | 42.80 |
| | Text-only | Multi-step w/o history | Acc. | 34.00 | 0.00 | 0.00 | 0.00 | 0.00 | 0.00 | 5.70 |
| | | | Comp. | 43.20 | 4.50 | 0.60 | 0.80 | 0.20 | 0.00 | 8.20 |
| | | | Eff. | 16.10 | 3.40 | 2.50 | 94.00 | 0.40 | 0.60 | 19.50 |
| | | Multi-step w/ history | Acc. | 26.00 | 0.00 | 0.00 | 0.00 | 0.00 | 0.00 | 4.30 |
| | | | Comp. | 32.80 | 4.00 | 0.30 | 0.80 | 1.00 | 0.00 | 6.50 |
| | | | Eff. | 8.60 | 3.40 | 2.30 | 95.60 | 0.50 | 0.50 | 18.50 |
| | | One-step | Acc. | 10.00 | 2.00 | 2.00 | 0.00 | 0.00 | 0.00 | 2.30 |
| | | | Comp. | 24.80 | 6.20 | 2.00 | 6.50 | 3.00 | 8.20 | 8.50 |
| | | | Eff. | 33.70 | 55.50 | 64.80 | 30.40 | 37.50 | 4.50 | 37.70 |
| | | Average | Acc. | 14.00 | 0.00 | 1.33 | 0.67 | 0.00 | 0.33 | 2.72 |
| | | | Comp. | 25.32 | 7.33 | 1.20 | 3.42 | 1.92 | 3.20 | 7.08 |
| | | | Eff. | 23.17 | 37.77 | 23.00 | 76.52 | 17.68 | 2.85 | 30.17 |
| | | Multi-step w/o history | Acc. | 4.00 | 0.00 | 0.00 | 0.00 | 0.00 | 0.00 | 0.70 |
| | | | Comp. | 10.00 | 5.80 | 0.00 | 1.80 | 3.00 | 0.00 | 3.40 |
| | | | Eff. | 2.70 | 21.40 | 1.70 | 35.60 | 17.40 | 0.40 | 13.20 |
| | | Multi-step w/ history | Acc. | 2.00 | 0.00 | 0.00 | 0.00 | 0.00 | 0.00 | 0.30 |
| | | | Comp. | 8.50 | 6.50 | 0.00 | 2.00 | 3.50 | 0.00 | 3.40 |
| | | | Eff. | 2.00 | 0.00 | 1.40 | 23.50 | 22.10 | 0.40 | 8.20 |

Image-text

| | | | | | | | | | |
|---|---|---|---|---|---|---|---|---|---|
| GPT-4o mini | Image-text | One-step | Acc. | 0.00 | 0.00 | 0.00 | 0.00 | 0.00 | 0.00 | 0.00 |
| | | | Comp. | 20.80 | 8.20 | 0.00 | 3.00 | 6.80 | 1.00 | 6.60 |
| | | | Eff. | 34.80 | 64.70 | 58.20 | 32.80 | 18.50 | 2.50 | 35.20 |
| | | Multi-step w/o history | Acc. | 12.00 | 0.00 | 0.00 | 0.00 | 0.00 | 0.00 | ,2.0 |
| | | | Comp. | 22.00 | 7.00 | 0.30 | 0.20 | 1.80 | 0.00 | 5.20 |
| | | | Eff. | 7.80 | 8.30 | 1.60 | 95.30 | 3.50 | 0.40 | 19.50 |
| | Text-only | Multi-step w/ history | Acc. | 14.00 | 0.00 | 0.00 | 0.00 | 0.00 | 0.00 | 2.30 |
| | | | Comp. | 23.80 | 8.80 | 0.00 | 1.20 | 1.50 | 0.00 | 5.90 |
| | | | Eff. | 8.40 | 4.90 | 1.80 | 86.10 | 2.50 | 0.30 | 17.30 |
| | | One-step | Acc. | 2.00 | 0.00 | 2.00 | 0.00 | 0.00 | 2.00 | 1.00 |
| | | | Comp. | 26.20 | 6.50 | 2.00 | 4.80 | 4.20 | 9.80 | 8.90 |
| | | | Eff. | 27.70 | 52.40 | 70.80 | 36.00 | 42.70 | 11.20 | 40.10 |
| | Average | | Acc. | 5.67 | 0.00 | 0.33 | 0.00 | 0.00 | 0.33 | 1.05 |
| | | | Comp. | 18.55 | 7.13 | 0.38 | 2.17 | 3.47 | 1.80 | 5.57 |
| | | | Eff. | 13.90 | 25.28 | 22.58 | 51.55 | 17.78 | 2.53 | 22.25 |
| | Image-text | Multi-step w/o history | Acc. | 0.00 | 0.00 | 0.00 | 2.00 | 0.00 | 0.00 | 0.30 |
| | | | Comp. | 10.00 | 5.80 | 0.00 | 7.20 | 0.20 | 0.00 | 3.90 |
| | | | Eff. | 25.10 | 33.60 | 2.00 | 86.80 | 8.40 | 5.50 | 26.90 |
| | | Multi-step w/ history | Acc. | 0.00 | 0.00 | 0.00 | 2.00 | 0.00 | 0.00 | 0.30 |
| | | | Comp. | 8.00 | 6.20 | 0.00 | 11.20 | 0.20 | 0.00 | 4.30 |
| | | | Eff. | 37.00 | 37.30 | 2.80 | 49.10 | 9.00 | 3.60 | 23.10 |
| | | One-step | Acc. | 28.00 | 2.00 | 4.00 | 4.00 | 0.00 | 4.00 | 7.00 |
| | | | Comp. | 55.00 | 5.50 | 4.00 | 13.80 | 6.50 | 46.60 | 21.90 |
| | | | Eff. | 51.30 | 63.40 | 60.20 | 52.50 | 26.40 | 36.90 | 48.40 |
| | Text-only | Multi-step w/o history | Acc. | 14.00 | 0.00 | 0.00 | 0.00 | 0.00 | 0.00 | 2.30 |
| | | | Comp. | 18.80 | 6.80 | 2.30 | 0.20 | 1.20 | 0.00 | 4.90 |
| | | | Eff. | 4.40 | 4.60 | 1.60 | 92.40 | 1.80 | 0.60 | 17.60 |
| | | Multi-step w/ history | Acc. | 14.00 | 0.00 | 0.00 | 0.00 | 0.00 | 0.00 | 2.30 |
| | | | Comp. | 21.20 | 6.20 | 1.40 | 0.00 | 2.50 | 0.00 | 5.20 |
| | | | Eff. | 4.60 | 4.00 | 1.70 | 98.80 | 1.60 | 0.40 | 18.50 |
| | | One-step | Acc. | 28.00 | 0.00 | 18.00 | 2.00 | 0.00 | 0.00 | 8.00 |
| | | | Comp. | 41.20 | 10.00 | 28.00 | 11.50 | 8.20 | 1.60 | 16.80 |
| | | | Eff. | 37.50 | 61.70 | 76.80 | 41.30 | 30.90 | 3.80 | 42.00 |
| | | | Acc. | 14.00 | 0.33 | 3.67 | 1.67 | 0.00 | 0.67 | 3.37 |
| | | | Comp. | 25.70 | 6.75 | 5.95 | 7.32 | 3.13 | 8.03 | 9.50 |

| Model | Modality | Method | Metric | | | | | | | |
|---|---|---|---|---|---|---|---|---|---|---|
| | | Average | Eff. | 26.65 | 34.10 | 24.18 | 70.15 | 13.02 | 8.47 | 29.42 |
| Claude-3 Opus | Text-only | Multi-step w/o history | Acc. | 14.00 | 0.00 | 0.00 | 0.00 | 0.00 | 0.00 | 2.30 |
| | | | Comp. | 18.20 | 8.00 | 0.90 | 0.50 | 1.50 | 0.00 | 4.80 |
| | | | Eff. | 5.40 | 3.10 | 1.70 | 62.80 | 1.20 | 0.40 | 12.40 |
| | | Multi-step w/ history | Acc. | 14.00 | 0.00 | 0.00 | 0.00 | 0.00 | 0.00 | 2.30 |
| | | | Comp. | 19.20 | 7.20 | 0.60 | 0.00 | 1.50 | 0.00 | 4.80 |
| | | | Eff. | 4.50 | 3.60 | 1.90 | 62.00 | 1.30 | 0.60 | 12.30 |
| | | One-step | Acc. | 0.00 | 2.00 | 2.00 | 2.00 | 0.00 | 0.00 | 1.00 |
| | | | Comp. | 40.50 | 4.80 | 6.00 | 7.50 | 4.20 | 1.20 | 10.70 |
| | | | Eff. | 40.20 | 55.70 | 71.00 | 47.00 | 27.70 | 3.00 | 40.80 |
| | | Average | Acc. | 9.33 | 0.67 | 0.67 | 0.67 | 0.00 | 0.00 | 1.87 |
| | | | Comp. | 25.97 | 6.67 | 2.50 | 2.67 | 2.40 | 0.40 | 5.07 |
| | | | Eff. | 16.70 | 20.80 | 24.87 | 57.27 | 10.07 | 1.33 | 21.83 |
| GPT-4 Turbo | Text-only | Multi-step w/o history | Acc. | 14.00 | 0.00 | 0.00 | 0.00 | 0.00 | 0.00 | 2.30 |
| | | | Comp. | 18.20 | 8.00 | 0.90 | 0.20 | 1.50 | 0.00 | 4.80 |
| | | | Eff. | 5.30 | 3.10 | 1.70 | 61.80 | 1.20 | 0.40 | 12.20 |
| | | Multi-step w/ history | Acc. | 14.00 | 0.00 | 0.00 | 0.00 | 0.00 | 0.00 | 2.30 |
| | | | Comp. | 19.20 | 7.20 | 0.60 | 0.00 | 1.50 | 0.00 | 4.80 |
| | | | Eff. | 4.50 | 3.60 | 1.80 | 62.00 | 1.30 | 0.60 | 12.30 |
| | | One-step | Acc. | 0.00 | 2.00 | 2.00 | 2.00 | 0.00 | 0.00 | 1.00 |
| | | | Comp. | 35.00 | 5.00 | 2.00 | 7.00 | 4.20 | 1.20 | 9.10 |
| | | | Eff. | 38.70 | 56.30 | 69.00 | 48.60 | 29.80 | 3.80 | 41.00 |
| | | Average | Acc. | 9.33 | 0.67 | 0.67 | 0.67 | 0.00 | 0.00 | 1.87 |
| | | | Comp. | 24.13 | 6.73 | 1.17 | 2.40 | 2.40 | 0.40 | 6.23 |
| | | | Eff. | 16.17 | 21.00 | 24.17 | 57.47 | 10.77 | 1.60 | 21.83 |
| Open Source Model | | | | | | | | | | |
| | Image-text | Multi-step w/o history | Acc. | 2.00 | 0.00 | 0.00 | 0.00 | 0.00 | 0.00 | 0.33 |
| | | | Comp. | 15.20 | 6.50 | 0.00 | 1.00 | 0.00 | 0.00 | 3.78 |
| | | | Eff. | 22.50 | 25.10 | 8.20 | 11.30 | 0.00 | 0.00 | 11.18 |
| | | Multi-step w/ history | Acc. | 0.00 | 0.00 | 0.00 | 0.00 | 0.00 | 0.00 | 0.00 |
| | | | Comp. | 13.20 | 6.80 | 0.00 | 0.00 | 0.00 | 0.00 | 3.33 |
| | | | Eff. | 19.80 | 26.00 | 7.30 | 9.80 | 0.80 | 0.00 | 10.62 |
| | | One-step | Acc. | 0.00 | 0.00 | 0.00 | 0.00 | 0.00 | 0.00 | 0.00 |
| | | | Comp. | 16.80 | 4.50 | 1.20 | 0.00 | 2.50 | 0.00 | 4.17 |
| | | | Eff. | 25.20 | 22.60 | 33.80 | 5.30 | 19.50 | 0.00 | 17.73 |

| Model | Setting | Metric | | | | | | | |
|---|---|---|---|---|---|---|---|---|---|
| MiniCPM-V2.6 | Multi-step w/o history | Acc. | 4.00 | 0.00 | 0.00 | 0.00 | 0.00 | 0.00 | 0.67 |
| | | Comp. | 14.00 | 5.50 | 0.00 | 1.20 | 1.00 | 0.00 | 3.62 |
| | | Eff. | 16.30 | 27.70 | 3.20 | 12.10 | 1.20 | 0.00 | 10.08 |
| | Text-only Multi-step w/ history | Acc. | 0.00 | 2.00 | 0.00 | 0.00 | 0.00 | 0.00 | 0.33 |
| | | Comp. | 11.20 | 4.00 | 0.10 | 0.80 | 0.00 | 0.00 | 2.68 |
| | | Eff. | 13.30 | 17.80 | 1.60 | 5.50 | 0.00 | 0.00 | 6.37 |
| | One-step | Acc. | 0.00 | 0.00 | 0.00 | 0.00 | 0.00 | 0.00 | 0.00 |
| | | Comp. | 14.00 | 5.80 | 2.00 | 0.00 | 3.50 | 0.00 | 4.22 |
| | | Eff. | 24.10 | 57.20 | 30.00 | 0.00 | 20.00 | 0.00 | 21.88 |
| | Average | Acc. | 1.00 | 0.33 | 0.00 | 0.00 | 0.00 | 0.00 | 0.22 |
| | | Comp. | 14.07 | 5.52 | 0.55 | 0.50 | 1.17 | 0.00 | 3.63 |
| | | Eff. | 20.20 | 29.40 | 14.02 | 7.33 | 6.92 | 0.00 | 12.98 |
| Internvl2-8B | Image-text Multi-step w/o history | Acc. | 6.00 | 0.00 | 0.00 | 0.00 | 0.00 | 0.00 | 1.00 |
| | | Comp. | 11.00 | 4.50 | 0.10 | 0.00 | 0.00 | 0.00 | 2.60 |
| | | Eff. | 17.90 | 11.10 | 3.40 | 3.00 | 0.00 | 0.00 | 5.90 |
| | Image-text Multi-step w/ history | Acc. | 2.00 | 0.00 | 0.00 | 0.00 | 0.00 | 0.00 | 0.33 |
| | | Comp. | 10.00 | 5.50 | 0.00 | 0.00 | 0.00 | 0.00 | 2.58 |
| | | Eff. | 16.60 | 10.90 | 2.80 | 1.30 | 0.00 | 0.00 | 5.27 |
| | One-step | Acc. | 2.00 | 0.00 | 0.00 | 0.00 | 0.00 | 0.00 | 0.33 |
| | | Comp. | 14.20 | 5.00 | 0.00 | 0.80 | 0.00 | 0.00 | 3.33 |
| | | Eff. | 10.40 | 12.00 | 3.30 | 4.10 | 0.00 | 0.00 | 4.97 |
| | Text-only Multi-step w/o history | Acc. | 8.00 | 0.00 | 0.00 | 0.00 | 0.00 | 0.00 | 1.33 |
| | | Comp. | 11.80 | 4.00 | 0.00 | 0.00 | 0.00 | 0.00 | 2.63 |
| | | Eff. | 13.20 | 2.30 | 0.00 | 2.80 | 0.00 | 0.00 | 3.05 |
| | Text-only Multi-step w/ history | Acc. | 4.00 | 0.00 | 0.00 | 0.00 | 0.00 | 0.00 | 0.67 |
| | | Comp. | 9.20 | 5.80 | 0.00 | 0.00 | 0.00 | 0.00 | 2.50 |
| | | Eff. | 13.10 | 6.60 | 1.50 | 4.40 | 0.00 | 0.00 | 4.27 |
| | One-step | Acc. | 10.00 | 0.00 | 0.00 | 0.00 | 0.00 | 0.00 | 1.67 |
| | | Comp. | 15.80 | 7.20 | 0.00 | 0.00 | 0.00 | 0.00 | 3.83 |
| | | Eff. | 17.70 | 10.10 | 0.00 | 8.40 | 0.00 | 0.00 | 6.03 |
| | Average | Acc. | 5.33 | 0.00 | 0.00 | 0.00 | 0.00 | 0.00 | 0.89 |
| | | Comp. | 12.00 | 5.33 | 0.02 | 0.13 | 0.00 | 0.00 | 2.91 |
| | | Eff. | 14.82 | 8.83 | 1.83 | 4.00 | 0.00 | 0.00 | 4.91 |
| | | Acc. | 12.00 | 2.00 | 0.00 | 0.00 | 0.00 | 0.00 | 2.33 |
| | | Comp. | 18.50 | 7.80 | 0.30 | 2.00 | 0.20 | 0.00 | 4.80 |

| Model | Input | Method | Metric | | | | | | | |
|---|---|---|---|---|---|---|---|---|---|---|
| Internvl2-26B | Image-text | Multi-step w/o history | Eff. | 25.00 | 19.70 | 2.20 | 14.90 | 1.20 | 0.50 | 10.58 |
| | | Multi-step w/ history | Acc. | 8.00 | 0.00 | 0.00 | 0.00 | 0.00 | 0.00 | 1.33 |
| | | | Comp. | 19.00 | 8.00 | 0.30 | 2.50 | 1.50 | 0.00 | 5.22 |
| | | | Eff. | 23.10 | 17.00 | 1.90 | 12.10 | 1.20 | 0.00 | 9.22 |
| | | One-step | Acc. | 10.00 | 0.00 | 0.00 | 0.00 | 0.00 | 0.00 | 1.67 |
| | | | Comp. | 21.20 | 8.50 | 1.20 | 2.50 | 0.50 | 0.00 | 5.65 |
| | | | Eff. | 27.70 | 18.60 | 15.50 | 7.40 | 2.40 | 0.00 | 11.93 |
| | Text-only | Multi-step w/o history | Acc. | 10.00 | 0.00 | 0.00 | 0.00 | 0.00 | 0.00 | 1.67 |
| | | | Comp. | 20.20 | 9.50 | 0.00 | 1.80 | 0.00 | 0.00 | 5.25 |
| | | | Eff. | 21.80 | 19.80 | 2.50 | 16.00 | 0.00 | 1.20 | 10.22 |
| | | Multi-step w/ history | Acc. | 12.00 | 0.00 | 0.00 | 0.00 | 0.00 | 0.00 | 2.00 |
| | | | Comp. | 20.00 | 10.50 | 0.10 | 1.00 | 0.00 | 0.00 | 5.27 |
| | | | Eff. | 20.10 | 21.20 | 1.30 | 10.80 | 0.00 | 2.20 | 9.27 |
| | | One-step | Acc. | 14.00 | 0.00 | 0.00 | 0.00 | 0.00 | 0.00 | 2.33 |
| | | | Comp. | 22.00 | 8.00 | 0.00 | 0.80 | 0.50 | 0.00 | 5.22 |
| | | | Eff. | 25.40 | 14.10 | 43.30 | 14.90 | 1.60 | 1.00 | 16.72 |
| | Average | | Acc. | 11.00 | 0.33 | 0.00 | 0.00 | 0.00 | 0.00 | 1.89 |
| | | | Comp. | 20.15 | 8.72 | 0.32 | 1.77 | 0.45 | 0.00 | 5.23 |
| | | | Eff. | 23.85 | 18.40 | 11.12 | 12.68 | 1.07 | 0.82 | 11.32 |
| | Image-text | Multi-step w/o history | Acc. | 8.00 | 2.00 | 0.00 | 0.00 | 0.00 | 0.00 | 1.67 |
| | | | Comp. | 21.20 | 6.80 | 0.60 | 4.50 | 1.00 | 0.00 | 5.68 |
| | | | Eff. | 33.50 | 40.40 | 1.60 | 29.70 | 0.80 | 4.20 | 18.37 |
| | | Multi-step w/ history | Acc. | 10.00 | 0.00 | 0.00 | 0.00 | 0.00 | 0.00 | 1.67 |
| | | | Comp. | 18.80 | 7.00 | 0.30 | 6.50 | 0.00 | 0.00 | 5.43 |
| | | | Eff. | 31.00 | 18.40 | 1.20 | 24.30 | 0.00 | 3.00 | 12.98 |
| | | One-step | Acc. | 16.00 | 0.00 | 0.00 | 0.00 | 0.00 | 0.00 | 2.67 |
| | | | Comp. | 30.00 | 7.50 | 2.00 | 6.50 | 1.20 | 0.00 | 7.87 |
| | | | Eff. | 42.40 | 42.60 | 18.80 | 28.10 | 1.40 | 0.00 | 22.22 |
| | | Multi-step w/o history | Acc. | 4.00 | 2.00 | 0.00 | 0.00 | 0.00 | 0.00 | 1.00 |
| | | | Comp. | 19.20 | 9.80 | 0.00 | 1.20 | 0.00 | 0.00 | 5.03 |
| | | | Eff. | 27.20 | 41.10 | 0.80 | 20.90 | 0.00 | 2.00 | 15.33 |
| | | Multi-step w/ history | Acc. | 12.00 | 0.00 | 0.00 | 0.00 | 0.00 | 0.00 | 2.00 |
| | | | Comp. | 12.50 | 7.50 | 0.00 | 4.00 | 0.50 | 0.00 | 4.08 |
| | | | Eff. | 24.10 | 39.00 | 1.80 | 25.80 | 0.60 | 0.00 | 15.22 |
| | | | Acc. | 10.00 | 0.00 | 0.00 | 0.00 | 0.00 | 0.00 | 1.67 |

Text-only

| Model | Modality | Setting | Metric | | | | | | | Avg |
|---|---|---|---|---|---|---|---|---|---|---|
| | | One-step | Comp. | 23.80 | 9.50 | 4.00 | 11.20 | 0.00 | 0.00 | 8.08 |
| | | | Eff. | 29.90 | 44.70 | 55.40 | 38.40 | 2.40 | 0,6 | 34.16 |
| | | Average | Acc. | 10.00 | 0.67 | 0.00 | 0.00 | 0.00 | 0.00 | 1.78 |
| | | | Comp. | 20.92 | 8.02 | 1.15 | 5.65 | 0.45 | 0.00 | 6.03 |
| | | | Eff. | 31.35 | 37.70 | 13.27 | 27.87 | 0.87 | 1.84 | 18.82 |
| Internvl-Chat-v1.5 | Image-text | Multi-step w/o history | Acc. | 0.00 | 4.00 | 0.00 | 0.00 | 0.00 | 0.00 | 0.67 |
| | | | Comp. | 28.30 | 9.50 | 0.00 | 0.00 | 0.00 | 0.00 | 6.30 |
| | | | Eff. | 36.40 | 42.20 | 3.20 | 6.70 | 0.00 | 0.90 | 14.90 |
| | | Multi-step w/ history | Acc. | 2.00 | 0.00 | 0.00 | 0.00 | 0.00 | 0.00 | 0.33 |
| | | | Comp. | 29.20 | 8.00 | 0.60 | 0.00 | 0.00 | 0.00 | 6.30 |
| | | | Eff. | 38.80 | 39.30 | 3.10 | 3.20 | 0.00 | 0.90 | 14.22 |
| | | One-step | Acc. | 0.00 | 0.00 | 0.00 | 0.00 | 0.00 | 0.00 | 0.00 |
| | | | Comp. | 18.50 | 6.50 | 1.20 | 1.00 | 0.20 | 0.00 | 4.57 |
| | | | Eff. | 26.00 | 36.70 | 58.10 | 26.30 | 1.10 | 5.90 | 25.68 |
| | Text-only | Multi-step w/o history | Acc. | 2.00 | 0.00 | 0.00 | 0.00 | 0.00 | 0.00 | 0.33 |
| | | | Comp. | 27.00 | 7.00 | 0.60 | 0.00 | 0.20 | 0.00 | 5.80 |
| | | | Eff. | 34.50 | 27.60 | 4.20 | 3.00 | 0.40 | 0.50 | 11.70 |
| | | Multi-step w/ history | Acc. | 2.00 | 0.00 | 0.00 | 0.00 | 0.00 | 0.00 | 0.33 |
| | | | Comp. | 29.50 | 6.50 | 0.00 | 0.00 | 0.00 | 0.00 | 6.00 |
| | | | Eff. | 39.10 | 22.10 | 1.60 | 2.40 | 0.00 | 0.00 | 10.87 |
| | | One-step | Acc. | 4.00 | 0.00 | 0.00 | 0.00 | 0.00 | 0.00 | 0.67 |
| | | | Comp. | 13.50 | 4.80 | 4.00 | 1.80 | 1.00 | 0.00 | 4.18 |
| | | | Eff. | 23.90 | 38.70 | 59.50 | 33.30 | 2.40 | 5.80 | 27.27 |
| | | Average | Acc. | 1.67 | 0.67 | 0.00 | 0.00 | 0.00 | 0.00 | 0.39 |
| | | | Comp. | 24.33 | 7.05 | 1.07 | 0.47 | 0.23 | 0.00 | 5.53 |
| | | | Eff. | 33.12 | 34.43 | 21.62 | 12.48 | 0.65 | 2.33 | 17.44 |
| | Image-text | Multi-step w/o history | Acc. | 4.00 | 0.00 | 0.00 | 0.00 | 0.00 | 0.00 | 0.67 |
| | | | Comp. | 9.00 | 8.00 | 0.10 | 2.50 | 1.20 | 0.00 | 3.47 |
| | | | Eff. | 14.40 | 28.00 | 2.50 | 24.80 | 0.50 | 0.60 | 11.80 |
| | | Multi-step w/ history | Acc. | 2.00 | 0.00 | 0.00 | 0.00 | 0.00 | 0.00 | 0.33 |
| | | | Comp. | 8.50 | 6.00 | 0.60 | 1.00 | 0.20 | 0.00 | 2.72 |
| | | | Eff. | 13.40 | 26.60 | 1.70 | 25.00 | 0.60 | 0.00 | 11.22 |
| | | One-step | Acc. | 6.00 | 0.00 | 0.00 | 0.00 | 0.00 | 0.00 | 1.00 |
| | | | Comp. | 10.80 | 6.50 | 0.60 | 4.00 | 0.00 | 0.00 | 3.65 |
| | | | Eff. | 16.70 | 30.50 | 29.90 | 21.30 | 0.00 | 0.00 | 16.40 |

| Model | Mode | | Metric | | | | | | | |
|---|---|---|---|---|---|---|---|---|---|---|
| DeepSeek-VL | Multi-step w/o history | | Acc. | 0.00 | 2.00 | 0.00 | 0.00 | 0.00 | 0.00 | 0.33 |
| | | | Comp. | 8.00 | 6.80 | 0.00 | 0.80 | 0.50 | 0.00 | 2.68 |
| | | | Eff. | 11.50 | 32.00 | 2.40 | 3.30 | 1.10 | 0.00 | 8.38 |
| | Text-only | Multi-step w/ history | Acc. | 0.00 | 0.00 | 0.00 | 0.00 | 0.00 | 0.00 | 0.00 |
| | | | Comp. | 11.80 | 5.80 | 0.00 | 6.50 | 1.00 | 0.00 | 4.18 |
| | | | Eff. | 16.10 | 25.40 | 0.00 | 15.20 | 0.70 | 0.00 | 9.57 |
| | | One-step | Acc. | 0.00 | 0.00 | 0.00 | 0.00 | 0.00 | 0.00 | 0.00 |
| | | | Comp. | 12.00 | 7.50 | 0.00 | 4.00 | 0.00 | 0.00 | 3.92 |
| | | | Eff. | 17.60 | 27.00 | 15.50 | 35.30 | 0.00 | 0.00 | 15.90 |
| | Average | | Acc. | 2.00 | 0.33 | 0.00 | 0.00 | 0.00 | 0.00 | 0.39 |
| | | | Comp. | 10.02 | 6.77 | 0.22 | 3.13 | 0.48 | 0.00 | 3.44 |
| | | | Eff. | 14.95 | 28.25 | 8.67 | 20.82 | 0.48 | 0.10 | 12.21 |
| Cogvlm2-19B | Image-text | Multi-step w/o history | Acc. | 8.00 | 0.00 | 0.00 | 0.00 | 0.00 | 0.00 | 1.33 |
| | | | Comp. | 25.50 | 7.50 | 0.00 | 1.20 | 1.20 | 0.00 | 5.90 |
| | | | Eff. | 33.10 | 45.30 | 1.10 | 11.10 | 2.80 | 1.10 | 15.75 |
| | | Multi-step w/ history | Acc. | 4.00 | 0.00 | 0.00 | 0.00 | 0.00 | 0.00 | 0.67 |
| | | | Comp. | 23.80 | 8.50 | 0.00 | 0.80 | 1.00 | 0.00 | 5.68 |
| | | | Eff. | 33.20 | 47.70 | 0.90 | 14.10 | 2.10 | 0.70 | 16.45 |
| | | One-step | Acc. | 12.00 | 0.00 | 0.00 | 0.00 | 0.00 | 0.00 | 2.00 |
| | | | Comp. | 27.00 | 8.80 | 1.20 | 2.50 | 0.00 | 0.00 | 6.58 |
| | | | Eff. | 39.50 | 39.80 | 47.50 | 28.80 | 0.00 | 7.10 | 27.12 |
| | Text-only | Multi-step w/o history | Acc. | 8.00 | 2.00 | 0.00 | 0.00 | 0.00 | 0.00 | 1.67 |
| | | | Comp. | 24.50 | 6.00 | 0.60 | 3.00 | 0.00 | 0.00 | 5.68 |
| | | | Eff. | 33.80 | 30.50 | 3.60 | 14.60 | 0.00 | 0.00 | 13.75 |
| | | Multi-step w/ history | Acc. | 8.00 | 0.00 | 0.00 | 0.00 | 0.00 | 0.00 | 1.33 |
| | | | Comp. | 20.50 | 7.00 | 0.60 | 1.00 | 1.00 | 0.00 | 5.02 |
| | | | Eff. | 31.00 | 31.40 | 2.30 | 10.50 | 0.50 | 1.40 | 12.85 |
| | | One-step | Acc. | 10.00 | 2.00 | 0.00 | 0.00 | 0.00 | 0.00 | 2.00 |
| | | | Comp. | 26.00 | 7.50 | 0.60 | 10.50 | 1.20 | 0.00 | 7.63 |
| | | | Eff. | 37.20 | 40.20 | 61.10 | 36.80 | 5.10 | 7.80 | 31.37 |
| | Average | | Acc. | 8.33 | 0.67 | 0.00 | 0.00 | 0.00 | 0.00 | 1.50 |
| | | | Comp. | 24.55 | 7.55 | 0.50 | 3.17 | 0.73 | 0.00 | 6.08 |
| | | | Eff. | 34.63 | 39.15 | 19.42 | 19.32 | 1.75 | 3.02 | 19.55 |
| | | | Acc. | 12.00 | 4.00 | 0.00 | 0.00 | 0.00 | 0.00 | 2.67 |
| | | | Comp. | 40.50 | 8.80 | 0.10 | 4.50 | 0.50 | 0.00 | 9.07 |

| Model | Modality | Setting | Metric | | | | | | | |
|---|---|---|---|---|---|---|---|---|---|---|
| InternVL2-Llama3-76B | Image-text | Multi-step w/o history | Eff. | 47.20 | 33.60 | 4.20 | 17.80 | 1.50 | 1.00 | 17.55 |
| | | Multi-step w/ history | Acc. | 12.00 | 2.00 | 0.00 | 0.00 | 0.00 | 0.00 | 2.33 |
| | | | Comp. | 30.20 | 4.50 | 0.30 | 2.50 | 0.20 | 0.00 | 6.28 |
| | | | Eff. | 38.60 | 28.90 | 3.90 | 15.30 | 1.60 | 2.50 | 15.13 |
| | | One-step | Acc. | 18.00 | 0.00 | 0.00 | 0.00 | 0.00 | 0.00 | 3.00 |
| | | | Comp. | 42.20 | 4.80 | 0.60 | 1.20 | 1.00 | 0.00 | 8.30 |
| | | | Eff. | 50.90 | 55.40 | 42.30 | 50.00 | 8.60 | 9.90 | 36.18 |
| | Text-only | Multi-step w/o history | Acc. | 14.00 | 0.00 | 0.00 | 0.00 | 0.00 | 0.00 | 2.33 |
| | | | Comp. | 41.80 | 6.80 | 0.30 | 0.80 | 0.20 | 0.00 | 8.32 |
| | | | Eff. | 49.40 | 49.20 | 5.50 | 20.70 | 1.90 | 0.10 | 21.13 |
| | | Multi-step w/ history | Acc. | 10.00 | 0.00 | 0.00 | 0.00 | 0.00 | 0.00 | 1.67 |
| | | | Comp. | 37.00 | 9.50 | 0.00 | 1.20 | 0.50 | 0.00 | 8.03 |
| | | | Eff. | 41.00 | 52.50 | 4.90 | 55.30 | 10.00 | 12.10 | 29.30 |
| | | One-step | Acc. | 14.00 | 4.00 | 0.00 | 0.00 | 0.00 | 0.00 | 3.00 |
| | | | Comp. | 18.80 | 9.00 | 2.00 | 4.50 | 1.00 | 0.00 | 5.88 |
| | | | Eff. | 29.00 | 52.00 | 47.70 | 49.00 | 9.90 | 10.10 | 32.95 |
| | Average | | Acc. | 13.33 | 1.67 | 0.00 | 0.00 | 0.00 | 0.00 | 2.50 |
| | | | Comp. | 35.08 | 7.23 | 0.55 | 2.45 | 0.57 | 0.00 | 7.65 |
| | | | Eff. | 42.68 | 45.27 | 18.08 | 34.68 | 5.58 | 5.95 | 25.38 |

Table 4: Results for all of the MLLMs

# E  CASE STUDY

## Sokoban | One-step | Image-text

**Prompt:**

You are a player of Sokoban game. And you will be given an image of a level of the Sokoban game.
Your task is to complete this level by outputting movement instructions based on this image one step at a time.

Objective: Move all boxes onto the designated storage locations (goals).

Rules:
  1. Movement: The player can move up (U), down (D), left (L), or right (R).
  2. Pushing Boxes: The player can push one box at a time by moving towards it. Boxes can only be pushed, not pulled.
  3. Grid Limitations: The player and boxes can only move into empty spaces. Walls and other boxes block movement.

Restrictions:
  a. A box cannot be pushed if there is another box or a wall directly behind it.
  b. The player cannot move through boxes or walls.

Illustration:
           1. dashed grid: dock
           2. yellow box: box on the dock (can also be pushed)
           3. brown box: box on the floor
           4. goal: push all the boxes onto the docks

Output Instructions:
  1. Provide movement instructions using only the 4 letters: "L" (left), "R" (right), "U" (up), "D" (down).
  2. Use JSON as your output format: {"output": "L"} or {"output": "R"} or {"output": "U"} or {"output": "D"}.

Please output only one step, and your output must be one of the following: {"output": "L"} or {"output": "R"} or {"output": "U"} or {"output": "D"} and not output anything else:

| Claude-3.5 Sonnet | Level 2 | GPT-4o |

**Claude-3.5 Sonnet:**

Output:
Based on the image provided, here's the solution to complete this Sokoban level:
{"output": "DLLDDRRUULLDLDDRRUU"}
This sequence of moves will push all the boxes onto the docks. Here's a breakdown of the moves:
  1. Move down and left to push the first box onto the dock.
  2. Move left again to get in position for the second box.
  3. Move down twice to get below the second box.
  4. Move right twice and up once to push the second box onto the dock.
  5. Move up again, then left twice to get in position for the third box.
  6. Move down twice to get below the third box.
  7. Move right twice and up twice to push the final box onto the dock.
This solution efficiently moves all three boxes onto their respective docks, completing the level.

**GPT-4o:**

Output:
To solve this Sokoban level, we need to move the boxes onto the dock (goal) spaces. Here is a step-by-step solution:
  1. Move the player to the right of the brown box.
  2. Push the brown box down onto the dock.
Here are the movement instructions:
  1. Move right: "R"
  2. Move down: "D"
  3. Move left: "L"
  4. Push the brown box down: "D"
So, the complete sequence of movements is:{"output": "RDLDD"}

**Analysis:**

Claude-3.5 Sonnet (Perceptual Error); GPT-4o (Perceptual Error):

Both models made perceptual errors, misidentified the locations of the boxes and docks.

**Correct Instruction**: "DUULLDDR"

Figure 7: A sample case of perceptual error. Sokoban — One-step — Image-text — Level 2.

# Hanoi | Multi-step | Text-only | w/ History

**Prompt:**

You are a player of Hanoi game. And you will be given an dictionary representation of a level of the Tower of Hanoi game.
Please finish the Tower of Hanoi puzzle based on the dictionary representation provided.

You must follow the rules of Hanoi game:

1. There are 4 rods: A, B, C, D
2. And 5 disks: a, b, c, d, e; for size: a > b > c > d > e
3. Your task is to move all the disks to rod "D"
4. Only one disk can be moved at a time
5. Only the top disk can be moved
6. At no time should a large disk be placed on top of a small disk.

Instructions:

1. Use JSON as your output format: {"output": "{rod_x}{rod_y}"}, which means move the disk on rod_x to rod_y
2. You will obtain a multi-turn conversation. The conversation history provided below may be helpful to you.

Dictionary representation:

{ "A": ["a", "c"], "B": ["b"], "C": ["d", "e"], "D": []}}

Conversation history:

{conversation_history}

Please output only one step based on the given rules, conversation history, and dictionary representation, and your output must meet required format {"output": "{rod_x}{rod_y}"}. Please do not output anything else.

|  Claude-3.5 Sonnet | Level 13 | GPT-4o |

**Claude-3.5 Sonnet:**

Output: "CD", "CD", "CD", "CD", "CD", "CD", "CD", "CD", "CD", "CD", "CD", "CD", "CD", "CD", "CD", "CD", "CD", "CD", "CD", "CD", "CD", "CD", "CD", "CD", "CD", "CD", "CD", "CD", "CD", "CD", "CD", "CD", "CD", "CD", "CD", "CD", "CD", "CD", "CD", "CD", "CD", "CD", "CD", "CD", "CD", "CD", "CD", "CD", "CD", "CD".

Steps: 50

**GPT-4o:**

Output: "Ce", "C D", "C D", "C D", "C D", "C D", "Ce", "Ce", C D", "Ce", "Ce", "C D", "C D", "Ce", "Ce","Ce", "Ce","Ce", "Ce", "Ce", "CDA", "Ce", "C D", "C D", "Ce", "C D", "C D", "CDA", "C D", "C D", "C D", "Ce", "C D", "Ce", "C D", "C D", "Ce", "Ce","Ce", "C D", "CDA", "C D", "C D", "C D", "Ce", "C D", "C D", "C D", "C D", "Ce".

Steps: 50.

**Analysis**:

Claude-3.5 Sonnet (Textual Understanding Error):
This model misunderstood the data structure representation of the Tower of Hanoi.

GPT-4o (Textual Understanding Error):
This model not only failed to understand the text representation of the Tower of Hanoi, but also failed to understand the prompt's requirements for the output format, resulting in the output of illegal text: "Ce", "C D", "CDA".

(Note: In the text-only setting, we do not use any image as model input and the image is only used for display.)

**Correct Instruction**: "AB", "AD", "BA", "BD", "AD", "CB", "CD", "BD"

Figure 8: A sample case of textual understanding error. Hanoi — Multi-step — Text-only — With-history — Level 13.

# 8-queens | One-step | Image-text

**Prompt:**

This is an image of a level of the n_queens game.
Your task is to generate a list of coordinates to complete the n_queens problem on a board where the first queen is already placed.

Follow these rules:
Each queen must be placed in such a way that no two queens threaten each other.
       1. No two queens can share the same row.
       2. No two queens can share the same column.
       3. No two queens can share the same diagonal.

Note:
  1. An 8 x 8 chessboard with 8 queens.
      2. The coordinate range is from 0 to 7.
  2. The position of the first queen (red color) is already given, so do not include it in your answer.
  3. Your output should be in the JSON format: {"output": [[row_x1, col_y1], [row_x2, col_y2], ...]}. Each [row_x, col_y] means the coordinate you want to place your piece.
  4. If your chess piece violates the three rules, it will be ignored.

Your answer:

Claude-3.5 Sonnet           Level 8           GPT-4o

**Claude-3.5 Sonnet**:

Output:
Based on the image and the rules provided, here's a solution for the 8-queens problem with the first queen already placed at the bottom-left corner (7,0):

  {"output": [[0,4], [1,7], [2,5], [3,2], [4,6], [5,1], [6,3]]}

This solution ensures that:
  1. No two queens share the same row
  2. No two queens share the same column
  3. No two queens share the same diagonal

The first queen (in red) at position (7,0) is not included in the output as instructed. The provided coordinates will place the remaining 7 queens in a way that satisfies all the rules of the n-queens problem on an 8x8 chessboard.

**GPT-4o**:

Output:
To solve the n-queens problem for an 8x8 board with the first queen already placed at (0, 0), we need to place the remaining 7 queens such that no two queens threaten each other. Here is one possible solution:

  {"output": [ [0, 0],[1, 2],[2, 4], [3, 6], [4, 1],[5, 3],[6, 5], [7, 7]]}

This configuration ensures that no two queens share the same row, column, or diagonal.

**Analysis**:

Claude-3.5 Sonnet (Planning Error):
Coordinates [2,5] and [0,7] violate the rule that no two queens can appear on the same diagonal.

GPT-4o (Perceptual Error, Textual Understanding Error):
GPT-4o made two main mistakes:
1. Identified the position of the first queen as [0,0];
2. Output the coordinates of the first queen.

**Correct Instruction**: [[0,4], [1,1], [2,3], [3,6], [4,2], [5,7], [6,5]]

Figure 9: A sample case of three errors. 8-queens — One-step — Image-text — Level 8.

# Maze | Multi-step | Image-text | w/o History

**Prompt:**

You are a player of Maze game. And you will be given an image of a level of the Maze game.
Your task is to move from your current position through the floor to the destination.

Rules:

    1. Red area: your current position.
    2. Green area: destination.
    3. Black area: wall, unable to pass.
    4. White area: floor, able to pass.

Output Instructions:
    1. Provide movement instructions using only the 4 letters: "L" (left), "R" (right), "U" (up), "D" (down).
    2. Use JSON as your output format: {"output": "L"} or {"output": "R"} or {"output": "U"} or {"output": "D"}.
    3. Output only one move at a time, wait for confirmation before proceeding to the next step.

Please output only one step, and your output must be one of the following: {"output": "L"} or {"output": "R"} or {"output": "U"} or {"output": "D"} and not output anything else:

|  GPT-4o  |  Level 33  |  Claude-3.5 Sonnet  |

**GPT-4o:**

Outputs: "U", "R", "R", "R", "R", "R", "R","R", "R"

Steps: 9

**Claude-3.5 Sonnet**:

Outputs: "R", "R", "R","R", "U" , "U", "U" , "U",
"U" , "U", "U" , "U" , "U" , "U", "U" , "U", "U" ,
"U", "U", "U" , "U" , "U", "U" , "U", "U", "U", "U"
, "U", "U" , "U", "U" , "U", "U" , "U", "U" , "U",
"U" , "U", "U" , "U", "U" , "U", "U" , "U", "U" ,
"U", "U" , "U", "U" , "U".

Steps: 50

**Analysis:**

GPT-4o (Correct):
GPT-4o's first instruction was "U" (UP), which had no effect, but then GPT-4o output a different instruction, "R" (RIGHT), and repeated it, successfully passing the test.

Claude-3.5 Sonnet (Perceptual Error):
Claude-3.5 Sonnet's first four moves were correct, but when there were only four steps left to the destination, it kept outputting "U" (UP), running out of steps and failing.

**Correct Instruction**: "R", "R", "R", "R", "R", "R","R", "R".

Figure 10: A sample case of output comparison. Maze — Multi-step — Image-text — Without-history — Level 33.

# 15-Puzzle | Multi-step | Image-text | w/o History

**Prompt:**

You are a player of n_puzzle game. And you will be given an image of a level of the n_puzzle game. Please finish the n_puzzle based on the image provided.

Rules:
   1. The board is a square grid of size 4 * 4;
   2. The board contains 15 numbered tiles and one empty space;
   3. The goal is to rearrange the tiles so that they are in ascending order from the top left corner of the board;
   4. Valid moves are up, down, left, and right.

Output Instructions:
      1. Use JSON as your output format: {"output": number}.
      2. if the number is around the empty space, they will swap positions.

Please output only one step and your output must meet required format {"output": number}. Please do not output anything else.

Claude-3.5 Sonnet            Level 45            GPT-4o

**Claude-3.5 Sonnet:**

Outputs: 11, 10, 6, 5, 9, 13, 14, 15.

Steps: 8.

**GPT-4o:**

Output: 12, 8, 8, 8, 8, 8, 8, 8, 8, 8, 8, 8, 8, 8, 8, 8, 8, 8, 8, 8, 8, 8, 8, 8, 8, 8, 8, 8, 8, 8, 8, 8, 8, 8, 8, 8, 8, 8, 8, 8, 8, 8, 8, 8, 8, 8, 8, 8, 8, 8.

Steps: 50.

**Analysis:**

Claude-3.5 Sonnet (Correct):
Perfect, every step was exactly right.

GPT-4o (Perceptual Error):
Stuck at the number "8".

**Correct Instruction**: 11, 10, 6, 5, 9, 13, 14, 15.

Figure 11: A sample case of output comparison. 15-Puzzle — Multi-step — Image-text — Without-history — Level 45.

# Sudoku | One-step | Image-text

**Prompt:**

This is an image of a level of the Sudoku game.
Please finish the sudoku puzzle based on the image provided.

Rules:
1. In sudoku, each row, column, and 3x3 grid must contain all the digits from 1 to 9 exactly once without repeating.
2. You need to determine the number to fill in the blank based on the existing numbers.

instructions:
1. The top left number is at row 0, column 0; the bottom right number is at row 8, column 8.
2. Use JSON as your output format: {"output": {"{row}{column}": {number}, "{row}{column}": {number}, ...}}.
3. The range of {row} and {column} are 0-8, the range of {number} is 1-9.

Your answer:

Claude-3.5 Sonnet — Level 2 — GPT-4o

**Claude-3.5 Sonnet**:

Output:Based on the provided Sudoku puzzle image and the rules of Sudoku, I've solved the puzzle. Here's the completed solution in the requested JSON format:

{"output":
{"00": 1, "01": 6, "02": 2, "03": 4, "04": 7, "05": 8, "06": 9, "07": 3, "08": 5,
"10": 8, "11": 7, "12": 4, "13": 5, "14": 3, "15": 9, "16": 1, "17": 6, "18": 2,
"20": 9, "21": 3, "22": 5, "23": 2, "24": 6, "25": 1, "26": 8, "27": 7, "28": 4,
"30": 6, "31": 2, "32": 8, "33": 9, "34": 4, "35": 7, "36": 3, "37": 5, "38": 1,
"40": 7, "41": 4, "42": 9, "43": 1, "44": 5, "45": 3, "46": 6, "47": 2, "48": 8,
"50": 3, "51": 5, "52": 1, "53": 8, "54": 2, "55": 6, "56": 7, "57": 4, "58": 9,
"60": 4, "61": 9, "62": 3, "63": 6, "64": 1, "65": 5, "66": 2, "67": 8, "68": 7,
"70": 5, "71": 1, "72": 6, "73": 7, "74": 8, "75": 2, "76": 4, "77": 9, "78": 3,
"80": 2, "81": 8, "82": 7, "83": 3, "84": 9, "85": 4, "86": 5, "87": 1, "88": 6}}

This solution completes the Sudoku puzzle while adhering to the rules that each row, column, and 3x3 grid contains all digits from 1 to 9 exactly once without repetition.

**GPT-4o**:

Output:

{"output":
{"00": 1, "01": 6, "02": 2, "03": 7, "04": 8, "05": 9, "06": 3, "07": 5, "08": 4,
"10": 7, "11": 4, "12": 5, "13": 3, "14": 9, "15": 1, "16": 6, "17": 2, "18": 8,
"20": 9, "21": 3, "22": 5, "23": 2, "24": 6, "25": 1, "26": 8, "27": 7, "28": 4,
"30": 6, "31": 2, "32": 1, "33": 4, "34": 7, "35": 3, "36": 5, "37": 9, "38": 8,
"40": 7, "41": 4, "42": 9, "43": 1, "44": 3, "45": 6, "46": 2, "47": 8, "48": 5,
"50": 3, "51": 1, "52": 8, "53": 2, "54": 5, "55": 7, "56": 4, "57": 9, "58": 6,
"60": 4, "61": 3, "62": 6, "63": 5, "64": 1, "65": 2, "66": 7, "67": 8, "68": 9,
"70": 5, "71": 6, "72": 7, "73": 8, "74": 2, "75": 4, "76": 9, "77": 3, "78": 1,
"80": 2, "81": 8, "82": 7, "83": 3, "84": 9, "85": 4, "86": 1, "87": 6, "88": 5 }}

**Analysis**:

Claude-3.5 Sonnet (Correct):
The answer is exactly right.

GPT-4o (Perceptual Error):
GPT-4o can accurately identify which numbers exist in each row, but cannot recognize the poistions of blank.

**Correct Instruction**: "00":1, "03":4, "10":8, "32":8, "33":9, "44":5, "50":3, "51":5, "61":9, "71":1.

Figure 12: A sample case of output comparison. Sudoku — One-step — Image-text — Level 2.

