# OpenReview forum: "ING-VP: MLLMs Cannot Play Easy Vision-based Games Yet"
_ICLR.cc/2025/Conference — Submitted to ICLR 2025_

### Official Review · Reviewer_A4Je · 2024-11-01

**Soundness:** 3
**Presentation:** 3
**Contribution:** 2
**Rating:** 5
**Confidence:** 2

**Summary:**

The main contribution of this paper is the development of ING-VP, an interactive game-based vision planning benchmark specifically designed to evaluate the spatial reasoning and multi-step planning abilities of multimodal large language models (MLLMs). ING-VP introduces six distinct games with straightforward rules but cognitively challenging layouts, like Sokoban, Maze, and Sudoku, to test models across various reasoning and interaction settings. The benchmark assesses critical capabilities such as spatial imagination, stepwise reasoning, and action efficiency, and is evaluated with multiple state-of-the-art MLLMs, exposing their significant limitations in spatial reasoning and planning.

**Strengths:**

1. Comprehensive Evaluation Framework: ING-VP provides a unique, systematic way to analyze multi-step planning and spatial reasoning, filling a gap in existing benchmarks that typically lack interactive or complex reasoning tasks.
2. Varied Task Settings: By testing models under different conditions (e.g., with or without historical context, one-step vs. multi-step reasoning), ING-VP offers nuanced insights into MLLMs' strengths and weaknesses.
3. Benchmark for Future Advancements: ING-VP sets a new standard for evaluating MLLMs, promoting the development of models that better mimic human cognitive functions.

**Weaknesses:**

1. Limited Task Diversity: The six games, while varied, may not fully capture the range of planning and reasoning tasks MLLMs might encounter in real-world applications.

2. Current Model Limitations: The paper notes that current MLLMs struggle even on simple tasks, suggesting that the benchmark might be too challenging without significant advancements in multimodal reasoning models. The paper does not provide or discuss on potential direction for current weakness of MLLM on vision-based games.

3. No Difficulty Grading: The lack of difficulty grading may hinder the benchmark’s ability to track progress across varying model capabilities or subtle improvements in reasoning skills.

**Questions:**

1) Is there any potential direction to improvement MLLM's ability on vision-based games? If not, please explain the reason.

2) Is it possible to generate more task with wider diversity automatically?

---

### Official Review · Reviewer_57qw · 2024-11-03

**Soundness:** 2
**Presentation:** 3
**Contribution:** 3
**Rating:** 6
**Confidence:** 3

**Summary:**

The research paper investigates the multistep planning and reasoning capabilities of Multimodal Large Language Models (MLLMs). It uses six different games to evaluate the MLLMs on the planning capabilities. The study explores the models' abilities in both image-only and text-only formats and assesses performance across single-step and multi-step scenarios, with conditions both including and excluding historical context (with-history vs. without-history). They have evaluated several open-source and closed-source state-of-the-art models, the results highlight that the performance of MLLMs remains subpar across various tasks. This is an important area of research, as it highlights key limitations and potential areas for improvement in the development of MLLMs.

**Strengths:**

1) The paper addresses the important problem of long-term reasoning and multistep planning tasks for multimodal LLMs. They have described several multistep reasoning tasks based on different games.
2) The chosen game environments provide a good foundation for evaluating these capabilities of the MLLs.
3) The authors evaluated several state-of-the-art models, both open-source and closed-source, revealing that MLLMs still perform below expectations across these tasks. This research is significant as it highlights key limitations and opportunities for improvement in the development of MLLMs.

**Weaknesses:**

Some additional experiments that will improve the quality of the paper:

1) The authors appear to use either image-only or text-only inputs for the model. Conducting experiments with both image and text as inputs could further enrich the analysis and provide insights into the model's ability to handle multimodal information effectively. It would be good to include such experiments.

2) For the text-only setup, did you use a language-only model, or did you use a vision-language model (VLM) without image tokens? It would be valuable to conduct experiments in both scenarios to compare their effectiveness.

3) In Fig. 3, it is unclear whether the planning errors are calculated only in the absence of perceptual errors, or if they include cases both with and without perceptual errors. Could you clarify this distinction?

4) Including experiments to emphasize the following aspects would add valuable insights to the study:

- Prompt sensitivity. Assessing whether the model’s performance on planning tasks is sensitive to variations in prompt wording could reveal if prompt adjustments lead to improved outcomes.
- Prompt non-determinism. Evaluating the model's responses under different hyperparameters, such as temperature, would demonstrate the stability of its outputs.
- Repeatability. Running experiments multiple times and reporting the mean and standard error would provide a clearer picture of performance consistency and reliability.

5) It would be helpful to highlight the model's behavior across all games, from very simple settings to more complex ones, to show how it performs at each level of complexity.

6) Finally it would be very useful if the authors further provide any additional insights into why the model performs so poorly on these planning and reasoning tasks?

**Questions:**

Answering questions mentioned in the weakness section could be useful.

---

### Official Review · Reviewer_gj26 · 2024-11-05

**Soundness:** 2
**Presentation:** 3
**Contribution:** 2
**Rating:** 3
**Confidence:** 4

**Summary:**

This work introduces ING-VP, an evaluation benchmark for multi-modal large language models (MLLMs) featuring six visual-based games. These games are designed to assess multi-step reasoning and spatial awareness, offering a unique perspective for evaluating MLLMs. Experimental results indicate that both advanced closed-source and open-source LLM models currently achieve low accuracy on this benchmark.

**Strengths:**

1. This paper presents an interesting approach by using visual games to evaluate the multi-step reasoning and spatial awareness abilities of MLLMs.
2. This paper provides an insightful analysis of the reasons why MLLMs frequently fail in these games.
3. The paper is well-structured and easy to follow.

**Weaknesses:**

1. The paper does not clearly differentiate itself from previous work. For instance, some studies evaluate MLLMs in agent-based environments, such as GUI and robotics, assessing multi-step reasoning and spatial abilities.
2. One of my concerns is the limited scope of the scenarios. The six visual games are pretty specific, and the perceptual or reasoning abilities they test may not adequately represent MLLMs' capabilities in a broader range of real-world scenarios.
3. I noticed the paper does not include advanced prompt-based methods like Chain-of-Thought (CoT) prompting, which is a standard approach in most benchmarks.

**Questions:**

Please see the weaknesses.

---

### Official Review · Reviewer_K8zR · 2024-11-06

**Soundness:** 3
**Presentation:** 2
**Contribution:** 1
**Rating:** 3
**Confidence:** 4

**Summary:**

This paper proposed ING-VP, a new benchmark that can be used to test the zero-shot performance of MLLMs on visual interactive games. They separate the benchmark into 6 games and then test 15 open- and closed-source MLLMs on them. Results show that most of MLLMs perform bad in most of sub-tasks, Claude-3.5 Sonnet only achieve 3.37% overall accuracy, GPT-4o achieve 2.75%.

**Strengths:**

1. This paper has a good dataset contribution, the new dataset can be one part of the benchmark groups for vision-language models.

2. The idea of this paper is quite clear and easy to follow. We all agree that MLLMs and Vision Language Model can not solve many vision tasks and it is useful to mention they are struggling with these visual interactive games.

3. This paper tests a large group of open-sourced and closed-sourced models.

**Weaknesses:**

1. The motivation of your paper is limited, as it has been proposed by many papers working on MLLMs for visual games and puzzle games, such as [1],[2],[3] (And if you search key words [game, puzzle, VQA, VLM] in some database, you can find there are more works in this topic). I agree your benchmark may have some advantages (like grouping some tasks together) and may be better than these papers in different aspects, you should make a comparison and highlight these differences. And also I don't think grouping them together in a benchmark is a big contribution.

2. It is worth highlighting why these tasks are important for MLLMs. For example, as you showed in section 3.3, all these tasks can be transformed into a deterministic environment with predefined states, actions, so why not use RL in these tasks? Even if your goal is testing the zero-shot performance of MLLMs, it is still weird to us, because MLLMs are not pretrained with these tasks. If you let a human to answer it, humans still need additional training before start doing these tasks.

3. What might be the upper bound performance of these tasks? Human's performance or RL models' performance? I feel that you should present them in your paper.

4. Can you give us an example of the Image-text and text-only testing sample in Table 1? What is the environment prompt you are using to interact with the test-only LLMs? And how about if you apply CoT for these tasks? The current version is quite unclear.

5. If it is possible, maybe you could try the RLHF to finetune some open-sourced MLLM for these tasks. It can be used as the baseline because all of your tasks are a deterministic environment and can be easily adapted into a reward function.

References:
[1] Chia, Yew Ken, et al. "PuzzleVQA: Diagnosing Multimodal Reasoning Challenges of Language Models with Abstract Visual Patterns." arXiv preprint arXiv:2403.13315 (2024).

[2] Ghosal, Deepanway, et al. "Are Language Models Puzzle Prodigies? Algorithmic Puzzles Unveil Serious Challenges in Multimodal Reasoning." arXiv preprint arXiv:2403.03864 (2024).

[3] Kraaijveld, Koen, et al. "COLUMBUS: Evaluating COgnitive Lateral Understanding through Multiple-choice reBUSes." arXiv preprint arXiv:2409.04053 (2024).

**Questions:**

See Weaknesses for more details.

I have put my questions together with some parts in the weakness, so you don't need to reply them again here.

----------------------------------------------------------------------------------------------------------

Post rebuttal:

I take a quick view of the modified version of the authors' paper. It solves part of my concerns. However, the main issue is still not solved. There are too many MLLM benchmarks nowadays, but most of them are not useful and most of them are overlapped to each others.

---

### Meta-Review · Area_Chair_YEgo · 2024-12-10

**Metareview:**

The paper received ratings of 6, 5, 3, and 3, which are primarily below the acceptance threshold. Reviewers raised several concerns, including insufficient motivation, lack of differentiation from previous work, the need for additional experiments, and limited task diversity. The authors did not respond to the reviewers' feedback, leaving several questions unanswered. Hence, the AC recommends rejection.

**Additional Comments On Reviewer Discussion:**

There was no discussion since the authors did not provide a rebuttal.

---

### Decision · Program_Chairs · 2025-01-22

Reject